

# Measurement Report: New insights into the boundary layer revolution impact on new particle formation characteristics in three megacities of China

**Hancheng Hu[1], Yidan Zhang [1], Yuting Li[1], Dongyang Pu [1], Hao Wu[1,*]**

1College of Electronic Engineering, Chengdu University of Information Technology, Chengdu 610225

* Correspondence: wuhao@cuit.edu.cn; 86 18613878407

**Abstract:** New particle formation (NPF) events contribute more than 60% of ultrafine particles particularly in the boundary layer. This study retrieved the particle number size distribution and the NPF parameters and their relationship with planetary boundary layer height (PBLH) evolution, as well as the air mass back trajectories during NPF events in three Chinese cities: Beijing, Guangzhou, and Shanghai. Furthermore, all NPF events has been classified into three types: new particles grow rapidly during the initial rise of the boundary layer in Type I events, while they grow after the boundary layer reaches a certain height (above 800 m) in Type II events, and the shrinkage cases are the Type III. The results show that particle growth dynamics categorized into distinct types demonstrate that sustained particle growth predominantly occurred under conditions of stable and elevated PBLH. Survival parameters ranged from 13.1 to 115.9 in Beijing, 9.0 to 110.2 in Guangzhou, and 8.4 to 25.6 in Shanghai. Specifically, Type I events were associated with survival parameters between 14.0 and 45.2. A significant negative correlation is observed between survival parameters and PBLH ( $R^2 = 0.2$ in Beijing, $R^2 = 0.02$ in Guangzhou, and $R^2 = 0.99$ in Shanghai, respectively). The main source of Aitken mode transport to Beijing is from Mongolia region. In Guangzhou, the contribution mainly comes from Jiangxi and Fujian provinces located in the northeast, while in Shanghai, the source lies to the northwest. This research provides valuable insights into developing strategies to manage the atmospheric environment.

Keywords: new particle formation; planetary boundary layer height; parameter correlation; backward trajectory.

## 1. Introduction



New particle formation (NPF) is the process by which detectable nanoscale clusters are
formed through heterogeneous nucleation of supersaturated gas molecules in the
atmosphere (Kanawade et al., 2022). This phenomenon occurs with remarkable
regularity in the atmosphere and culminates in the formation of significant nucleation
mode aerosol particles through condensation and coagulation. These newly formed
particles evolve over spatial scales of hundreds of kilometers and timescales of 1–2
days, potentially affecting the global climate as cloud condensation nuclei (CCN) (Zhao
et al., 2017; Kerminen et al., 2018). NPF events occur frequently in various atmospheric
environments (Li et al., 2023b), playing a major role in the evolution of particle number
concentration. The processes of early nucleation, continuous growth and occurrence
frequency are critical issues in aerosol formation and Ultrafine particle source research.
Exploring its relevant mechanisms has significant impacts on regional environmental
quality, climate, and human health (Kulmala et al., 2022). Currently, many methods
have been used to identify the growth characteristics and development mechanisms of
NPF (Hu et al., 2016; Chan et al., 2020), such as field observations (Kerminen et al.,
2018), model simulations and smoke chamber simulations (Chu et al., 2022). Among
these, model simulations and smoke chamber simulations are usually employed to study
the chemical mechanisms of NPF.Especially in the boundary layer, most NPF are under
the control of the daily evolution of the planetary boundary layer height (PBLH) (Du et
al., 2025; Sun et al., 2015)so the influence should be further studied.
In China, research on NPF process parameters has been conducted through field
observations in multiple locations, including Beijing (Kulmala et al., 2021; Wu et al.,
2021b) and the Northwest Desert region (Shengjie et al., 2001). A long-term (373 days)
observation of NPF in urban Beijing examined the formation and growth of sub-3 nm
particles. It found that a sluggish growth rate (GR) and the presence of pristine
background aerosols lead to a reduced survival rate of newly formed particles (Deng et
al., 2021a). Moreover, ultrafine particles (UFP, diameter < 100nm) are greatly affected
by the condensation sink (CS), with 95% of NPF events occurring when CS < 0.03 s$^-$



[1] (Deng et al., 2021b). The impact of PBLH on NPF is complex and significant, involving multiple factors such as meteorological conditions, precursor gas concentrations, and particle growth mechanisms. Various atmospheric conditions influence not only the frequency of NPF but also its development. High relative humility could be responsible for the enhanced growth (Du et al., 2025) and at the higher temperature the cluster evaporation markedly slows NPF (Li et al., 2023a). Furthermore, due to the lower availability of condensable species, NPF events occurring under cleaner atmospheric conditions exhibit lower GR of newly formed particles (Bousiotis et al., 2018). The relationship between PBLH and NPF is intricate and warrants further in-depth research.

As a key process of air pollution, NPF can change boundary layer structure directly or indirectly by influencing the surface energy balance through radiation effects (Myhre et al., 2013). Shen et al. (2016) reported the diurnal variation in NPF number concentration observed at Mount Tai is primarily governed by the evolution of the boundary layer, while the seasonal variation is driven by both boundary layer evolution and atmospheric advection over Mount Tai. Deot et al. (2024) conducted measurements at different altitudes in Cyprus and found that higher PBLH are typically associated with stronger NPF events, which attributed to under higher boundary layer conditions, more precursor gases can accumulate at elevated temperatures and with lower CS rates. The boundary layer development can govern the capacity for atmospheric vertical diffusion, and changes in its height will directly affect the transport process of NPF, indicating a direct feedback system between NPF and the PBL. The influence of NPF on the PBLH varies with seasonal conditions: during colder periods (autumn and winter), lower PBLH correlates negatively with nucleation-mode particle (Nuc mode), while in warmer periods (spring and summer), lower PBLH correlates positively with NPF events (Blanco-Alegre et al., 2022). In warmer periods (spring and summer), higher PBLH supports NPF by enhancing vertical mixing and creating a cleaner atmosphere (Blanco-Alegre et al., 2022). Additionally, high aerosol concentrations in





the PBL can suppress NPF by reducing particle sources and increasing sinks, shifting
NPF occurrences to the lower troposphere (Quan and Jia, 2020). The flux of harmful
aerosol pollution in the boundary layer depends on atmospheric turbulent mixing,
which is closely related to the PBLH (Stjern et al., 2023). A high CS environment can
hinder the growth of new particles, while the boundary layer can affect CS, thereby
influencing NPF. The PBLH impacts the vertical distribution of particles (Wu et al.,
2021a). The PBLH influences the distribution and concentration of particles (Wang et
al., 2021), which in turn affects the CS by altering the surface area available for vapor
condensation (Sebastian et al., 2021).
However, long-term observations of NPF are rare in China, with only a few studies
documenting NPF observations spanning more than one year (Peng et al., 2017; Chu et
al., 2019). In this study, we utilized long-term observational datasets to analyze particle
variations, backward trajectories, and pollution distribution maps of megacities (Beijing,
BJ; Guangzhou, GZ; Shanghai, SH) in China during NPF periods. We use the same
setup and observation procedure around the cities. The effective observation days of BJ,
GZ, and SH were 408, 127, and 53 days, respectively. The back trajectory is estimated
from the Hybrid Single-Particle Lagrangian Integrated Trajectory (HYSPLIT) model
developed by the NOAA Air Resources Laboratory. This approach considers the
diversity of different natural atmospheric conditions and human activities, providing a
basis for clarifying the similarities and differences in the sources and transport
processes of particulate pollutants across different megacities regions in China
mainland.
The research methods and locations of the observation sites are presented in Chapter 2.
Chapter 3 discusses the occurrence frequency of NPF, the seasonal daily variations in
aerosol PNSD at the three sites, and the 48-hour backward trajectories of particulate
matter during NPF events. Additionally, we conducted research on the PBLH of the
three cities during NPF events. This study provides an important reference for



environmental management in China.
**2 Experiments and methods**
**2.1 Station location**
We set up an intensity Carbin installed muti particle instruments, move around three
observation sites to collect data. They are distributed separately in three major Chinese
cities, BJ, GZ, and SH. The scanning mobility particle sizer (SMPS) is used to measure
the particle sizes of atmospheric aerosols. The micro pulse lidar (MPL) can be
connected to computers to monitor PBLH data in real-time via specialized software.
The equipment setup is shown in Fig. 1a and 1b.

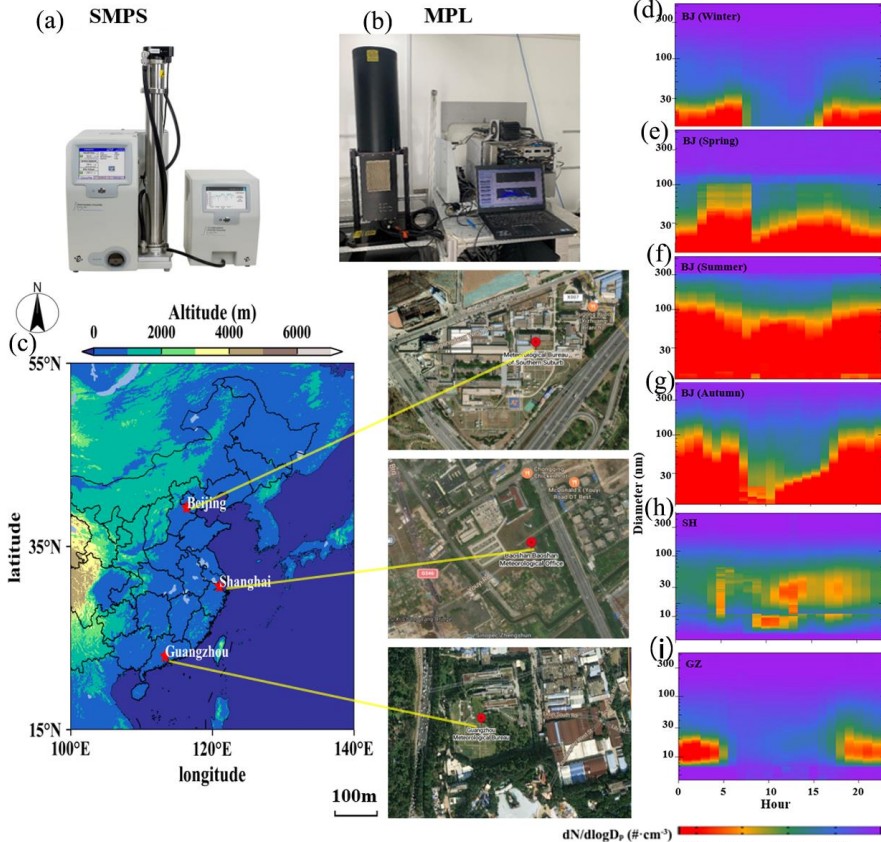




**Fig. 1** The instrument: (a) SMPS and (b) MPL. (c) Depiction of instruments, geographical location diagram and surroundings of three observation sites in BJ, GZ and SH. (d-g) The diurnal variation of PNC during observations in BJ in different seasons from July 2017 to October 2019, (h) in SH from April to June 2020, and (i) in GZ from November 2019 to March 2020, respectively.

The observation site was established at Beijing Observatory (116.35° E, 39.2° N, 44 m a.s.l.), located along the Fifth Ring Road in the Daxing District's Old Palace area, the southern suburb of BJ, within the North China Plain. The terrain is elevated in the northwest and lower in the southeast, surrounded by mountains to the west, north, and northeast. The site experiences a temperate, semi-humid to semi-arid monsoon climate. The surrounding traffic network is intricate, with significant traffic congestion. The diurnal variation of PNSD in different seasons in BJ is significant (Fig. 1d–Fig. 1g). In winter (December, January, and February), From 8:00 to 15:00 local time (LT), the PNC of smaller than 30 nm particles remained below 1,000 cm$^{-3}$, which was one of the reasons why NPF was lower than in other seasons. In summer, the PNC of smaller than 50 nm particles stayed above 4,000 cm$^{-3}$. These UFPs are typically primary particles that facilitate the growth of new particles in the atmosphere.

The site in GZ is the observation field of Guangzhou Meteorological Bureau (113°E, 23°N, 11 m a.s.l), 10 km from the center of GZ. An urban trunk road lies 500 m north of the site and another 100 m west, with the Chimelong Tourist Area and several industrial parks in proximity. GZ is in the central part of Guangdong Province and the northern edge of the PRD, with terrain that slopes from northeast to southwest. GZ is one of China's earliest open coastal cities, serving as a critical hub for transport and logistics. The city enjoys a subtropical monsoon climate with significant maritime influences. The particles were mainly concentrated at night, while the air was much cleaner during daylight hours. Starting from 17:00 LT, particles in the 25-50 nm range began to increase, likely due to the influence of land-sea circulation, which recirculated 25-50 nm particles. This phenomenon will be explained about the PBLH in the



following sections.
The observation site of SH is set at Baoshan Meteorological Bureau Observatory (121°
E, 30.6° N, 2 m a.s.l). SH is situated in the alluvial plain of the YRD in Eastern China,
bordered by Asia to the west and the Pacific Ocean to the east, with a population
exceeding 24 million. The city experiences a subtropical monsoon climate,
characterized by ample sunlight and precipitation. Compared to PNSD in BJ and GZ,
the air in SH was the cleanest, with consistently low average PNC throughout the day
(shown in Fig. 1h). PNC in SH was primarily concentrated during daylight hours,
differing from their distribution in GZ. The PNC for particles smaller than 30 nm
increased mainly between 21:00 LT and 2:00 LT.
**2.2 Equipment and Dataset**
The Scanning Mobility Particle Sizer (SMPS) is used to measure the particle sizes of
atmospheric aerosols. The Micro Pulse Lidar (MPL) can be connected to computers to
monitor PBLH data in real-time via specialized software. The equipment setup is shown
in Fig. 1a and 1b. The measurements of aerosol particle number size contribution
(PNSD) were made with SMPS. It measures particle size contribution from 11.3 nm to
552.3 nm in 1 min intervals and a Nano-SMPS has been added from December 2, 2018,
to obtain the 2-40 nm PNSD. The autonomous MPL operates at 532 nm in both parallel
and perpendicular polarizations, providing backscattered radiation profiles at a
temporal resolution of 10–30 s and a vertical resolution of 30 m (Roldán-Henao et al.,
2024). PBLH was derived from MPL measurements, except for SH during 1–29 April
2020 due to the Lidar misfunction during these period, where it was retrieved from
ERA5 reanalysis data.
The effective PNSD observation datasets used in the analysis include 408 days (July
2017–October 2019) in BJ, 127 days (November 2019–March 2020) in GZ, and 53
days (April 2020–May 2020) in SH, respectively. The observation datasets used to plot





the PNSD are three-dimensional, including datetime, particle size range (Dp, the
particle diameter), and particle number concentration (in dN/dlogDp) for the
corresponding particle size. After obtaining the original data, we need to eliminate the
abnormal values. In this study, the hourly average of real-time data was used to
construct the PNSD plot, The characteristics of aerosol PNSD can determine its CS,
GR, P value and physical in the atmosphere. The sources of aerosol particles can be
identified through the analysis of their size distribution.

**182 2.3 Identify NPF events and calculation of NPF parameters**

In general, specific criteria are applied to identify NPF events. A typical NPF event
should exhibit the following three characteristics, often summarized as a "banana"
shape (Maso et al., 2005). First, there is an increase in nucleation-mode PNC (particles
with 10–25 nm diameter). Second, new nucleation-mode particles develop
independently and persist for several hours. Finally, nucleation particles show a
continuously increasing trend over 3 hours (Maso et al., 2005). If a process meets the
above criteria, it is classified as an NPF event, and that day is defined as an NPF day.
If there was no clear growth of newly formed particles or the particle growth is
intermittent in time, the day was classified as undefined. Days without a burst of
nucleation-mode particles and subsequent growth were considered non-event days
(Non_NPF).
Calculated GR during NPF using the geometric mean diameters of the fitted particle
size distributions:

$$GR_{\Delta D_p} = \frac{dD_p}{dt} = \frac{dD_P}{\Delta t}$$


where $D_p$ is the measured diameter during NPF at time $t$ . We represent GR by
calculating the linear fitting slope of the geometric mean diameter of particles within a





certain time period (Casquero-Vera et al., 2023).
CS represents the loss of gaseous vapors due to their condensation onto aerosol particles.
It is positively correlated with the collision sink and does not directly affect PNC. The
formula is as follows (Kulmala et al., 2012):

$$CS = 4\pi D \int_0^{d_{p}max} {}_m d_p N_{d_p} dd_p = 4\pi D \sum_{d_p} \beta_{m,d_p} d_p N_{d_p}$$


Where $D$ is the diffusion coefficient of the condensed gas, $\beta_m$ is the correction
coefficient.
**2.4 Survival Parameter**
Survival parameter is crucial for determining the occurrence of NPF events, as it
represents the fraction of newly formed particles that survive the transition from a
smaller to a larger diameter. (Tuovinen et al., 2022). It can be used to predict the
dimensionless P of pollutant gas and provide guidance during periods of particulate
pollution (Kulmala et al., 2017). The formula of the dimensionless P is as follows:

$$P = \frac{CS'}{GR'}$$


Where the $CS' = CS / (10^{-4} s^{-1})$, and $GR' = GR / (1\ nm\ h^{-1})$. Both CS and GR are key
parameters calculated from PNSD. Generally, high CS values tend to suppress NPF, but
the survival probability can vary significantly due to environmental conditions and
particle interactions (Tuovinen et al., 2022). It is significant for investigating the
pollution source to research the P, as well as the impact of meteorological conditions
and seasonal changes of P value.
**2.5 Backward trajectories**





This study used global reanalysis of meteorological data to calculate the pollutant
source and air mass transport trajectories at three sites. We used a method combining
the Potential Source Contribution Function (PSCF) and Concentration-Weighted
Trajectory (CWT) in the HYSPLIT model to identify pollutant sources reaching
observation sites and assess their impact on NPF. PSCF shows the contribution from
potential source areas based on the conditional probability of pollution trajectories in
each grid but does not indicate the pollution level in the study area. The $ij_{th}$ component
of a PSCF field given as follows (Zong et al., 2018):
$$PSCF_{ij} = m_{ij} / n_{ij}$$

where $n_{ij}$ is the total number of endpoints that fall in the $ij_{th}$ cell, and $m_{ij}$ is the number
of endpoints in that plot for which the readings exceed a user defined threshold standard.
Note that cells with few endpoints can result in high uncertainty in the PSCF method.
Thus, to move these high uncertainties, an arbitrary weight function $W(n_{ij})$ is
multiplied into the PSCF value:
$$W_{ij} = \begin{cases} 1.0, \ N_{ij} > 3N_{Ave} \\ 0.70, 1.5N_{Ave} < N_{ij} < 3N_{Ave} \\ 0.42, N_{Ave} < N_{ij} < 1.5N_{Ave} \\ 0.17, 0 < N_{ij} < N_{Ave} \end{cases}$$

**3 Result and discussion**
**3.1 Frequency of NPF events**
As discussed in Sect. 2, NPF events were identified using the methodology and criteria
outlined by (Kulmala et al., 2012). The frequency of NPF events and average PBLH at
the three measurement sites are illustrated in Fig. 2. During the observation period, all
NPF events occurred during the daytime growth. The average PBLH was calculated





from 8:00 to 18:00 LT, which also could be the NPF window 9:00-15:00 LT.
We have found 74 NPF events during 408 effective observation days in BJ, from July
2017 to October 2019 (details in Table 1 which lists all NPF days during the observation
period in BJ, GZ, and SH). The NPF frequency in BJ was 17.8%, with 12 events
occurring in summer (June, July, and August). The summer months had the lowest
occurrence, indicating that factors such as meteorological conditions and precursor
availability, alongside PBLH, potentially influence NPF events (Deng, 2020) . NPF
events occurred predominantly in spring and autumn months, with the notably highest
frequencies observed in March (25.9%) and October (23.3%). Temperature emerged as
the primary factor driving seasonal variations in the frequency and intensity of NPF in
BJ (Li et al., 2023a). The monthly mean PBLH for Beijing averaged around 643 m,
peaking at approximately 833 m and displaying a positive correlation with the NPF
frequencies. For certain months, especially November and December, the substantial
number of missing observations restricts the ability to comprehensively assess the
influence of the boundary layer on NPF events. A more detailed discussion will be
provided in the following sections.
A total of 14 NPF events were recorded during 127 effective observation days in GZ
from November 2019 to March 2020. The mean NPF frequency was relatively low,
averaging around 10.0%, with frequencies ranging from 3.4% to 20.0%. The mean
PBLH was approximately 512 m, peaking at approximately 586 m, reinforcing the link
between atmospheric mixing conditions and particle formation processes.
A total of 10 NPF events, with a frequency of 18.87%, were observed in SH over 53
days from April to June 2020. The mean PBLH in SH was approximately 677 m,
reaching a maximum near 741 m at noon, with the highest values coinciding notably
with high NPF frequencies, particularly in April.



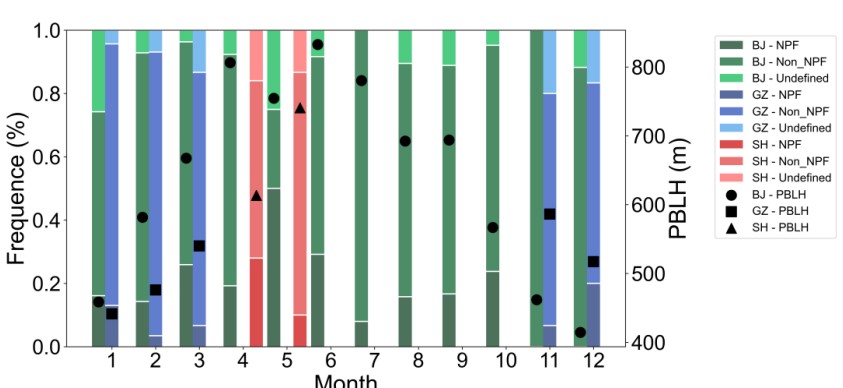

**Fig. 2.** Frequency of NPF events and average PBLH in BJ, GZ, and SH. The circle denotes PBLH of BJ, the square denotes PBLH of GZ, and the triangle denotes PBLH of SH.

**Table 1**. New particle formation events at BJ, GZ and SH.

| site | Month | PBLH (m) | NPF days | Non_NPF days | Undefined days | Missing days |
|------|-------|----------|----------|--------------|----------------|--------------|
| BJ | Jan | 458.6719 | 5 | 18 | 8 | 0 |
| BJ | Feb | 581.5099 | 4 | 22 | 2 | 0 |
| BJ | Mar | 667.5479 | 7 | 19 | 1 | 4 |
| BJ | Apr (in 2 years) | 806.1902 | 5 | 19 | 2 | 35 |
| BJ | May | 754.4776 | 2 | 1 | 1 | 26 |
| BJ | Jun (in 2 years) | 832.5667 | 7 | 15 | 2 | 30 |
| BJ | Jul (in 2 years) | 780.34 | 7 | 30 | 0 | 32 |
| BJ | Aug (in 2 years) | 692.309 | 5 | 14 | 2 | 31 |
| BJ | Sep | 693.769 | 3 | 13 | 2 | 13 |
| BJ | Oct | 566.9179 | 5 | 15 | 1 | 10 |
| BJ | Nov (in 2 years) | 461.7263 | 0 | 2 | 0 | 58 |
| BJ | Dec (in 2 years) | 414.4698 | 0 | 15 | 2 | 45 |
| GZ | Jan | 441.4076 | 3 | 19 | 1 | 8 |
| GZ | Feb | 476.0462 | 1 | 26 | 2 | 0 |
| GZ | Mar | 539.949 | 2 | 24 | 4 | 1 |
| GZ | Nov | 586.0955 | 2 | 22 | 6 | 0 |





| | | | | | | |
|------|------|----------|---|----|---|---|
| GZ | Sep | 517.0407 | 6 | 19 | 5 | 1 |
| SH | Apr | 613.2995 | 7 | 15 | 4 | 5 |
| SH | May | 740.7455 | 3 | 23 | 4 | 1 |

## 3.2 PBLH under different events

The relationship between PBLH and NPF is further explored with a temporal resolution approach. The observation period was classified into NPF days and Non_NPF days, and Fig. 3 reveals clear distinctions in the diurnal variations of PBLH between NPF and Non_NPF days. Prior to sunrise (0:00-6:00 LT), PBLH remained at 150–200 m. With the onset of surface heating after 6:00 LT, the PBLH grown rapidly, especially on NPF days, reaching roughly 500 m by 9:00 LT compared to approximately 390 m on Non_NPF days. After 9:00 LT, the solar radiation that the surface receives increases, making PBLH rise gradually (Zheng et al., 2017). On NPF days, the maximum average PBLH in BJ was 1165 m, whereas on Non_NPF days, it peaked at approximately 962 m around 14:00 LT, marking a difference of about 203 m. In the late afternoon (16:00–18:00 LT), the PBLH values declined as insolation wanes. In GZ, the boundary layer remained relatively elevated on Non_NPF days, often exceeding 600 m even during nighttime. Vehicular emissions outside the observation sites contribute to an "aerosol greenhouse effect", enhancing nighttime thermal retention (Miao et al., 2019). Meanwhile, the urban heat island effect intensifies surface thermal turbulence, promoting the nocturnal rise of the boundary layer (Huang and Bai, 2023; Zhang et al., 2024). Across all three sites, the PBLH on NPF days was 100–200 m higher than on Non-NPF days. This difference was consistently observed across BJ, GZ, and SH, suggesting that enhanced convective mixing and reduced particle CS under high PBLH conditions provide favorable conditions for nucleation and particle growth.





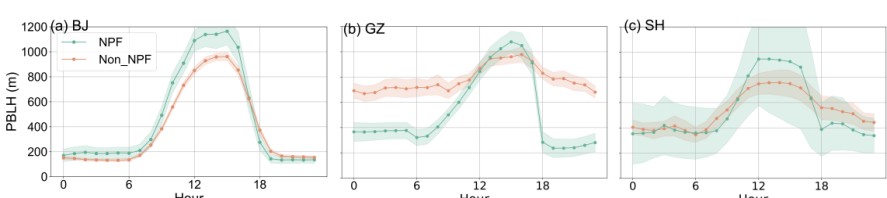

291

**Fig. 3.** The average of PBLH during NPF events and Non_NPF events at (a) BJ, (b) GZ, and (c) SH.

**3.3 Evolution Patterns and Particle Growth**

Within the dataset comprising 71 NPF events, four instances were observed to occur
concurrently with the boundary layer's ascent, classified as Type I. A second pattern,
designated as Type II, involves NPF events manifesting as atmospheric turbulence
intensifies, coinciding with boundary layer elevations exceeding 800 m Fig. 4a
illustrates a representative case of Type I on 24 August 2017, where at approximately
10:30 LT, the PBLH commenced its ascent from 450 meters. Concurrently, the particle
number concentration of nucleation mode particles (Nuc mode) surged from 294,644
$cm^{-3}$ to 503,055 $cm^{-3}$, indicating the initiation of an NPF event synchronized with the
development of the boundary layer. Conversely, a Type II event was characterized by a
rapid increase in Nuc at 11:30 LT, accompanied by the emergence of a distinct new
particle mode, indicative of the onset of NPF at this time. The PBLH increased from 22
m at 7:00 LT to approximately 1,000 m by 11:30 LT. Fig. 4c and 4d depict the temporal
correlation between average PBLH and Vehicular emissions outside the observation
sites contribute to an "aerosol greenhouse effect," enhancing nighttime thermal
retention. Meanwhile, the urban heat island effect intensifies surface thermal turbulence,
promoting the nocturnal rise of the boundary layer. for the respective event types. The
t represents the relative time at which the NPF event occurs, and t+1 refers to one
hour after the onset of the NPF event. Comparative analysis reveals that Type I events
predominantly occur when the PBLH is around 400 m, suggesting that initial boundary
layer development can serve as a catalyst for NPF. In contrast, Type II events are
initiated when the PBLH exceeds 800 m, implying a requirement for a more mature



boundary layer to facilitate nucleation processes. The gradual increase in PBLH leads
to a significant reduction in near-surface aerosol concentrations and facilitates the
accumulation of precursor vapors within a larger volume, thereby decreasing
condensation loss and creating favorable conditions for Nuc (Hao et al., 2018; Rose et
al., 2021; Zhang et al., 2024). As the PBLH approaches its maximum, photochemical
reactions become most intense, resulting in peak concentrations of oxidized precursor
gases (Rose et al., 2021). This fulfills the critical conditions for nucleation and
subsequently triggers NPF events. Meanwhile, the concentration of low-volatility
oxygenated organic aerosol is positively correlated with PBLH (Lin et al., 2021),
suggesting that in Type II events, NPF may be primarily driven by low-volatility
organics or $H_2SO_4$.

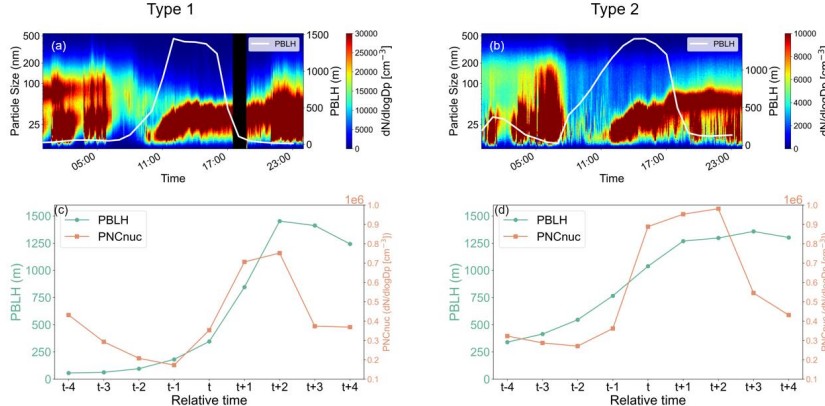


**Fig. 4.** Time series of aerosol particle number size distribution: (a) a case from Type I on 24 August
2017 and (b) a case from Type2 on 27 April 2018. Time series of averaged PBLH and Nuc mode
for (c) Type1 and (d) type2. The t denotes the relative time at which NPF occurs.
**3.4 Particle number size distribution of new particles formation events**
Fig. 5 presents a classification of NPF event types based on particle size distribution
evolution across three cities categorized as Type I, Type II, and Shrinkage events as





Type III. During the observation period, two inverse growth events were documented
in BJ, whereas only one such event was recorded in both GZ and SH. At the BJ, GZ,
and SH sites, Type I events typically started approximately two hours earlier than Type
II events. At BJ, nighttime bursts of polluted particles were observed on all 4 days;
however, the end times of these events could not be quantitatively determined. In
contrast, at both GZ and SH, the end times of Type I and Type II events were nearly
identical, coinciding with the rapid drop of the boundary layer to its minimum height.
The remaining two events, which occurred in BJ, involved particle growth from
approximately 100 nm at 17:00 LT to less than 25 nm during nighttime. In GZ, particles
formatted at 9:00 LT, reaching sizes up to 100 nm, followed by a reduction to below 30
nm after 15:00 LT. In SH, particle growth commenced at 10:00 LT, increasing from 10
nm to approximately 50 nm, before shrinking and reverting to nucleation mode by
15:00 LT. This investigation primarily examines the influence of PBLH on NPF events,
deliberately excluding the analysis of inverse growth mechanisms. Consequently, these
events are omitted from further detailed examination.

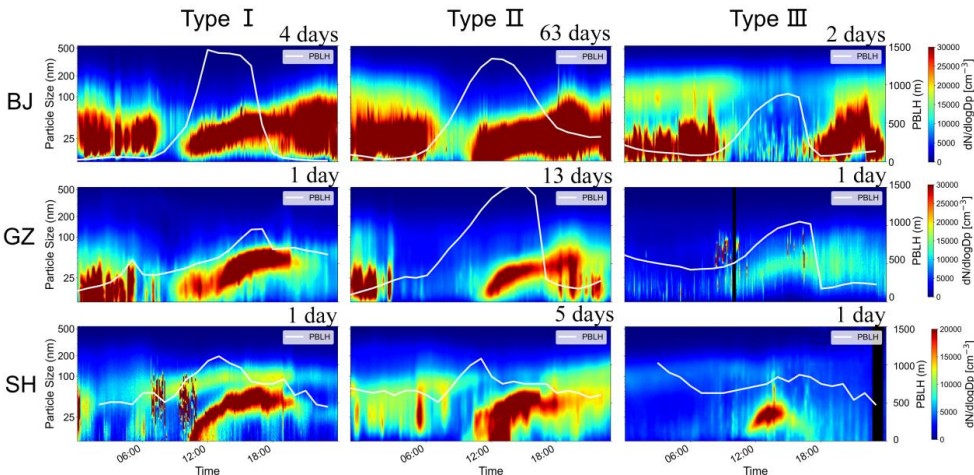

**Fig. 5.** Averaged aerosol particle number size distribution. PNSD of two types of NPF and growth,
and shrinkage days in BJ, GZ, and SH, respectively. Averaged PBLH add in the white line.





**3.5 The correlation between PBLH and P**

The relationship between PBLH and P during NPF events is illustrated in Fig. 6. In three cities, there was a negative correlation has been firstly built between PBLH and P parameters. The PBLH of BJ had the highest average with a value of 1010 m, and the P was highly variable, with a maximum of 153.92 and a minimum of 8.80. The P for BJ and GZ were similar. SH recorded the lowest average PBLH, with an average of 721.92 m. Among the three cities, SH was the cleanest, with P = 23.96. The higher influence of urban vehicular emissions in BJ and GZ may explain this difference. In contrast, SH's proximity to the sea led to higher relative humidity, which promoted NPF but also accelerated particle dissipation. However, the correlations in BJ and GZ were weak, with R² values of 0.21 and 0.024, respectively. In SH, the shorter observation period limited the number of captured events, showing a strong negative correlation in the available data, with an R² value of 0.99. As the PBLH increased, the P exhibited varying decreasing trends across the three cities. A high PBLH leads to a low P, which favors the occurrence of NPF events. The differences in correlation at different locations can be explained by site-specific factors. For instance, the GZ site is located near a major traffic artery, where particulate pollution from traffic sources weakens the correlation between P and PBLH. The decrease in the boundary layer provides a more stable environment for the particle accumulation, leading to a higher P value without suppose the NPF.



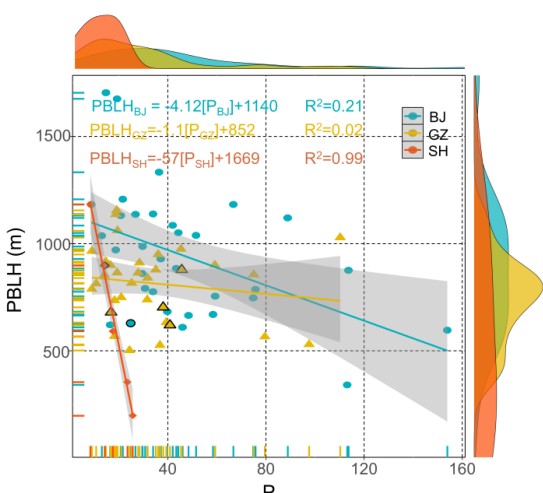

**Fig. 6** The correlation analysis between PBLH and P in (a) BJ, (b) GZ, and (c) SH. The circle

denotes BJ, the triangle denotes GZ, and the diamond denotes SH. The dots with black borders

represent Type I.

**3.6 The backward trajectories of particles during NPF events**

Nucleation-mode aerosols mainly originate from early nucleation and the subsequent

growth of NPF events. To track this, we meagered nucleation-mode concentration data

to 48-hour back trajectories during NPF periods. Overall, nucleation-mode particle

contributions to total concentration were greater in the northern parts of the study sites

compared to the southern parts. Fig. 8 presents the PSCF and frequency distribution of

NPF events across three cities.

Fig. 7a shows the PSCF results for the BJ site, indicating significant contributions from

Mongolia and cities within Anhui Province. In particular, pollution sources within

Mongolia and along the Mongolia-Russia border showed probabilities greater than 0.4.

The nucleation-mode particles were mainly from the northwest and the north of the

observation station. The pollutant source impacting BJ had the fastest transmission

speed within 48 hours and the broadest geographic reach, with primary contributions





originating from Russia and passing through Mongolia to BJ's northwest. Fig. 7b highlighted BJ and Zhangjiakou as the cities with the highest NPF event contribution among surrounding locations, where favorable geographic and environmental conditions promote the formation of particles smaller than 100 nm. The north-northwest of BJ exhibited the highest frequency of NPF events, exceeding 25%, while the area with the second-highest frequency extended approximately 420 km from the observation station, measured by latitude.

In GZ, as shown in Fig. 7c and 7d, pollutant contributions were primarily from Jiangxi and Fujian provinces to the northeast, with additional high-contribution areas extending eastward into Hubei Province. The distribution of atmospheric particle pollutants in this area is highly concentrated. The region with a high frequency (> 25%) of NPF events lies within 100 km to the northeast of the observation site, corresponding to the eastern portion of Guangdong Province.

For SH, pollutant contributions are mainly from the west-northwest, similar to BJ, although the overall pollutant contribution is lower than that for BJ. The contribution area spans Jiangsu Province and extends into southern Hebei Province. Additionally, there are notable contributions from the southwest, passing through Anhui and Jiangxi Provinces. Around SH, NPF event contribution levels are relatively high (approximately 10-25%), with the northern region showing the highest frequency, exceeding 25%. 10%-25% of the contribution came from the sea, indicating that marine circulation is one of the important pathways for the transport of new particles.





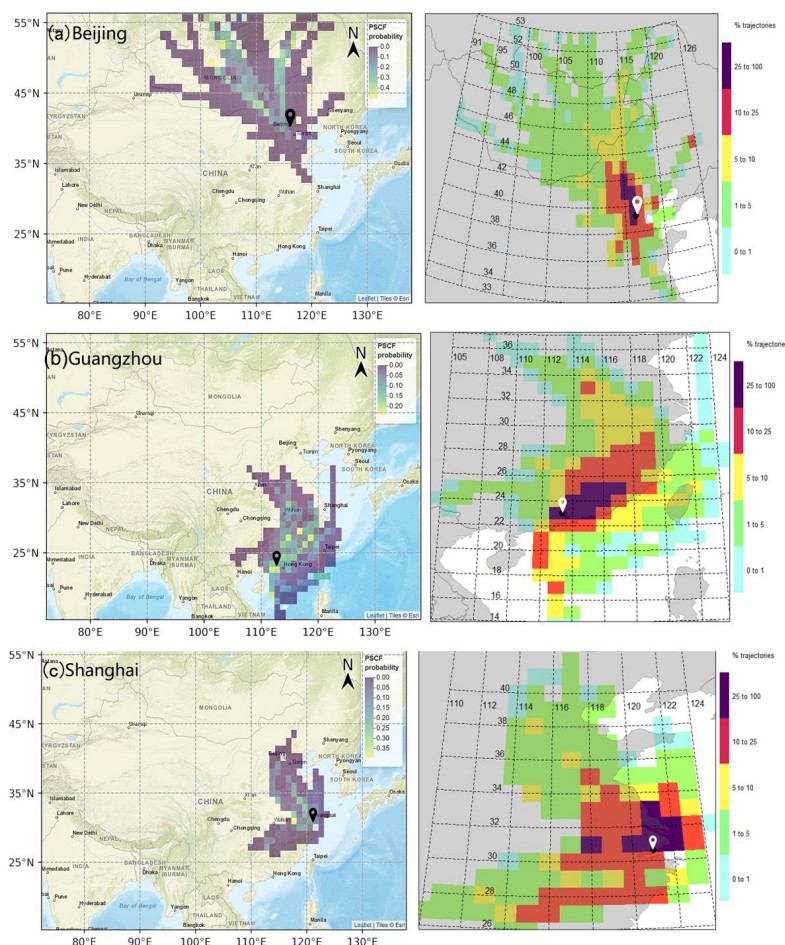

409

**Fig. 7** The 48h backward trajectory by using PSCF, and the map of NPF event contribution levels

when particle size is below 100nm in BJ, GZ, and SH during NPF days.

## 4 Conclusions

This study examined the relationship between PBLH and NPF events key parameters

in Chinese's megacities (BJ, GZ, and SH, respectively). During the observation period,

March and May in BJ exhibited the highest frequencies of NPF occurrence, accounting

for 25.9% and 23.8%, respectively. Diurnal analyses confirmed that NPF days were





consistently associated with significantly higher midday PBLH compared to Non_NPF
days. Across all three sites, the average PBLH on NPF days was 150–200 m higher than
on Non-NPF days, highlighting the importance of convective mixing and aerosol
dilution in promoting nucleation. We also identified two distinct mechanisms: We
identified two distinct mechanisms of NPF initiation: Type I and Type II. Type I refers
to events triggered during the initial rise of the boundary layer, where turbulent mixing
associated with PBLH development facilitates nucleation. Type II involves nucleation
that occurs only after the boundary layer reaches a certain height (>800 m). Correlation
analyses emphasized the boundary layer was a key factor in triggering NPF the level
especially at SH 0.99 This was particularly evident at SH, where the PBLH shows a
strong negative correlation with the P value. Indicating the vertical mixing process and
the development of the boundary layer has a dominant impact on the key parameter of
NPF events.
We obtained the 48h backward trajectories of particles at three sites during NPF by
using the HYSPLIT model. The main source of pollutant contribution in BJ is Mongolia
in the northwest direction. GZ's contribution source is distributed in Jiangxi and Fujian
Provinces in the northeast of the site, and SH's is in the northwest. The average
frequency level of NPF is over 25%, which is in the north of each site, indicating that
the overall level of air pollution in the north is higher than that in the south. Our finding
provides a brand-new insight into atmospheric turbulence and boundary layer
development could has dominate influence on the NPF and UFP formation mechanism,
which should not be ignored in further research.
**Data availability**
The measured data described in this manuscript can be accessed at the data repository maintained by
Mendeley Data. Doi: 10.17632/zpwjj5ymmp.1 (Hu, 2025)
**CRediT authorship contribution statement**



HH collected the resources, wrote, and finalized this paper. HH, YZ, TL, and DP analyzed the data and
generated the figures. HW planned the study, provided instruments and data, and discussed the results.
HH plotted the figures. HH, HW conducted the measurements. HW revised this paper.

**Declaration of competing interest**

The authors declare that they have no known competing financial interests or personal
relationships that could have appeared to influence the work reported in this paper.

**Acknowledgments**

This research was supported by the National Natural Science Foundation of China
(42105073), Protect of the Sichuan Department of Science and Technology
(2022NSFSC1074). It was also supported by the Key Laboratory of China
Meteorological Administration Atmospheric Sounding.

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
