# Peer review of "Measurement Report: New insights into the boundary layer"

_EGUsphere, 2025_

## Author Comment (AC1)

**Response to Reviewers' Comments**

Thank you very much for your letter and for the reviewers' comments regarding our manuscript entitled "Measurement Report: New insights into the boundary layer revolution impact on new particle formation characteristics in three megacities of China" (Manuscript ID: EGUSPHERE-2025-3637). These comments are all valuable and very helpful for improving our manuscript. We have addressed the reviewers' comments point-by-point and revised the manuscript accordingly. All changes made in the manuscript are highlighted in blue in the marked version of the manuscript. The detailed responses to the editor and reviewers' comments are shown as follows:

Reviewer Comment:

The authors mainly discuss the connection between PBLH on NPF via the decrease in condensation and coagulation sink in the morning due to the extending PBLH. This effect has been studied for over 20 years (Nilsson et al., 2001) by several groups and is widely accepted as one of the main reasons why NPF events are observed to take place on clear sky days, starting some hours after the sunrise, when the PBLH development starts. The other main reason is the important role of sulphuric acid, produced via photo-oxidation of $SO_2$, in NPF. The possible roles of decreasing RH and mixing of compounds in the mixing layer and the residual layer are also investigated. Many of these are not caused by increase of PBLH, but because of increase in solar radiation on the surface and the resulting increase in temperature. In this manuscript, the authors study the relations between BBLH and NPF occurrence only via the connection between PBLH and dimensionless parameter P (ratio of condensation sink and particle growth rate). If the aim is to use this parameter as the ratio between factors inhibiting

and enhancing NPF, the vapours responsible for the growth should be assumed to be the same that form the initial clusters in NPF. This is not likely, as the GR already in < 3nm is typically observed to be higher than sulphuric acid would produce. The authors do not present the connections between PBLH and CS or CS and NPF occurrence, even though they discuss these connections repeatedly. The determined GRs, or the size ranges they represent, are not shown and their reliability not discussed.

Response:

"The established connection between PBLH and NPF":

Thanks for the editor's comments. We acknowledge that the relationship between planetary boundary layer height (PBLH) and new particle formation (NPF), particularly in relation to the decrease in condensation and coagulation sinks, has been well-studied over the past two decades, with Nilsson et al., (2001) being a key reference in this area. We agree that this mechanism is widely accepted and serves as a fundamental explanation for the observed NPF events. However, our study seeks to build on this existing knowledge by focusing specifically on the role of the dimensionless survival parameter P (ratio of condensation sink (CS) to particle growth rate (GR)) in describing the interaction between PBLH and NPF. While the relationship between PBLH and NPF has been explored through several physical mechanisms, we aim to provide an understanding of this connection using a new framework.

Analysis of boundary layer and NPF development suggests that random and sudden mixing processes in a thermally unstable atmosphere favor particle nucleation (Bigg, 1997; Nilsson et al., 2001). Atmospheric stability influences the mixing of gases and particles (Alonso-Blanco et al., 2025). Aircraft measurements have shown that boundary layer development and vertical mixing facilitate the burst of ultrafine particles

in the residual layer (Platis et al., 2015), and are highly correlated with the occurrence of NPF events (Hao et al., 2018; Leino et al., 2019) .

**"The role of sulfuric acid and other factors in NPF":**

On the role of sulfuric acid and other factors in NPF: We concur with the reviewer that sulfuric acid plays a significant role in NPF, particularly through photo-oxidation of $SO_2$. Our manuscript does not intend to discount this established fact but focuses more specifically on the interactions between PBLH and the dimensionless survival parameter P. Our main objective in focusing on PBLH is to approach the analysis from an aerodynamic perspective. We aim to investigate whether changes in the PBL play a significant role in NPF, specifically in terms of their impact on CS and particles.

As pointed out by the reviewer, these NPF events are indeed influenced by factors such as solar radiation and temperature. We agree with the reviewer's opinion that radiation plays a significant role in NPF events. In response to the reviewer's comment, we downloaded 20 years of ground-based radiation observation data (Liu et al., 2022) from the Xianghe site to analyze the influence of radiation on NPF events. The results of our analysis will be presented in detail later. We will revise the manuscript to better emphasize the interplay of these factors and their contribution to NPF events.

For sulfur compounds and $SO_2$, we downloaded hourly data from the China National Environmental Monitoring Center and calculated $H_2SO_4$. We estimated the sulfuric acid proxy $[H_2SO_4]$ based on local solar radiation, $SO_2$ concentration, CS, and RH (Mikkonen et al., 2011).

$$proxy[H_2SO_4] = 8.21 \times 10^{-3} \times k \times radiation \times SO_2{}^{0.62} \cdot (CS \cdot RH)^{-0.13}$$

where $k$ is the temperature-dependent reaction-rate constant. The relative error between calculated sulfuric acid proxy and measured sulfuric acid concentration is estimated to be 42 % (Mikkonen et al., 2011; Xiao et al., 2015)

Due to the proximity of the Beijing site to a highway, pollutants accumulate at night, with particle sizes primarily concentrated in the ranges of 15-200 nm. In Phase 1, horizontal winds greater than 3 m/s diluted the pre-existing particles and further reduced the CS. After sunrise, with the increase in radiation, the PBLH began to rise from 130 m, destabilizing the boundary layer and enhancing vertical dilution, which led to the lowest CS value of 0.011 $S^{-1}$. This created favorable conditions for NPF. In this phase, PBLH shows a negative correlation with CS, indicating that as PBLH increases, CS decreases. This suggests that a higher PBLH is conducive to reducing CS. By 12:00, NPF events began to occur. In Phase 3, particles grew to over 100 nm. At this point, the PBL stabilized, with the PBLH decreasing to below 200 m. Both CO and CS began to rise, with nighttime traffic emissions contributing significantly to the particle number size distribution (PNSD) during this period.

[Figure]

Fig. R1. Evolutions of a typical regional NPF event (24 August 2017) and associated variables in Beijing. (a) 1-h average wind vector. Arrows represent the wind direction, and their lengths show the wind speed. (b) Time series of temperature, RH, PBLH, $SO_2$, $O_3$, CO, and CS. (c) The particle-number size distribution. The white dashed curve shows radiation. The dash box is the NPF window (11:00-19:00).

Fig. R2. shows the correlations between various factors on August 24, 2017. Radiation

exhibits a strong positive correlation with Nuc (11-25 nm) (p = 0.85), which is in line with the reviewer's observation of a significant relationship with radiation. However, PBLH also shows a strong correlation, with a notable positive correlation for particles in the 25-100 nm range (p = 0.74). Additionally, PBL has a clearing effect on particles larger than 100 nm, removing coarse particles and reducing CS. Radiation, on the other hand, shows almost no correlation with CS (p = 0.03), thus confirming that changes in PBL can lead to a lower CS. On this day, the correlation between 11-100 nm particles and both T and RH was less than 0.3. The NPF was primarily the result of the combined effects of radiation and PBLH. Radiation is associated with photochemical processes, while PBL reflects the aerosol dynamics within the PBLH.

[Figure]

Fig. R2. The Spearman correlation coefficient of various parameters.

"The assumption of vapors responsible for particle growth":

The reviewer raises an important point regarding the assumption that the same vapors responsible for particle growth are involved in the formation of initial clusters in NPF. We agree that this assumption may not always hold, especially considering the higher GR observed in particles smaller than 3 nm. In our research, we use the survival parameter P to more intuitively illustrate the impact of PBL on NPF, as P is one of the key factors in assessing NPF. We did not collect data for particles smaller than 3 nm; our primary concern is whether particles in the 11.3-25 nm range during NPF events can grow to over 100 nm. Additionally, when evaluating the impact of PBL on NPF, relying solely on CS and GR is insufficient, as pointed out by Cai et al., (2022).

"The connections between PBLH and CS, and the reliability of GR":

The reviewer rightly points out that we have not presented the direct connections between PBLH and CS or CS and NPF occurrence in the manuscript, despite discussing them multiple times. We will revise the manuscript to explicitly present these connections, with a clearer explanation of how PBLH influences CS and, in turn, the probability of NPF events. In addition, we also explored the relationship between PBLH and PNCnuc, using case studies to illustrate how boundary layer development influences CS, NPF events, and PNCnuc. As shown in Fig. R2, PBLH is negatively correlated with CS and simultaneously promotes the increase of PNCnuc, thereby enhancing the likelihood of NPF occurrence. This is further supported by the case shown in Fig. R5, which also demonstrates that while PBLH can influence CS, the occurrence of NPF depends on multiple contributing factors. This will help enhance the robustness of our findings.

Reviewer Comments:

The results on the connection between PBLH and NPF occurrence are first based on the mentioned positive correlation between monthly mean PBLH and NPF frequencies (lines 251-253). However, the highest values of PBLH are between April and July (Table 1), but the lowest NPF occurrence is during summer months (lines 245-246 and figure 2). This correlation should be represented and investigated more in detail.

Response:

We thank the reviewer for this valuable comment. Indeed, the apparent contradiction between the monthly trend of PBLH and the frequency of NPF events highlights the importance of examining their relationship with greater nuance.

The higher frequency of NPF events in Beijing during spring and autumn compared to summer is consistent with previous observational studies (Wu et al., 2021; Wu et al., 2007). This conclusion is based on observational data; however, the current analysis remains relatively superficial, which may lead to some misinterpretation regarding the relationship between PBL and NPF frequency. We will further elaborate on and revise this part in the updated manuscript. Of course, the lower NPF frequency in summer compared to spring and autumn should be examined in conjunction with other variables, such as radiation, $SO_2$, temperature, and relative humidity, for a more comprehensive analysis.

In Fig. R3c, all pollutant concentrations were normalized. During summer, $SO_2$ levels decreased by approximately 1.5 compared to winter, indicating a significant reduction in primary pollution and sulfur precursors. Although summer is characterized by strong photochemistry, elevated $O_3$ levels, and abundant OH production, the lack of $SO_2$ limits

the formation of sufficient sulfuric acid to trigger nucleation. Additionally, the average radiation in July dropped from around 420 to 320 W m$^{-2}$, likely due to the rainy season, which further reduced the production rate of $H_2SO_4$ (Dada et al., 2020). The high-humidity conditions in Summer also contribute to an increase in CS, which is unfavorable for NPF (Kulmala et al., 2013).

[Figure]

mean and 95% confidence interval in mean

Fig. R3. The variation of PBLH, radiation, $NO_2$, CO, $SO_2$, $O_3$, T, RH, and WS for all days in Beijing. The shading shows the 95 % confidence intervals of the mean.

Fig. R4 shows the variation of CS during summer. In June and July, CS values were relatively low, with average values of 0.038 $s^{-1}$ and 0.036 $s^{-1}$, respectively. However, in August, the average CS exceeded 0.2 $s^{-1}$. Despite the presence of a high PBLH, which typically leads to lower CS, CS remained elevated. Such high CS levels are unfavorable for NPF, which explains the lower frequency of NPF events observed in August. The correlations shown in Figure R4 (the right line) were calculated using the Spearman method to assess relationships among various factors. The correlation coefficients between CS and temperature, RH, wind speed, PBLH, and radiation were all below 0.2. It should be noted that these calculations were based on the full dataset, without accounting for factors such as multivariate interactions or temporal lags. Further analysis based on specific case studies will be presented in the following sections.

[Figure]

Fig. R4. The variation of CS for all days in Beijing (a) in June, (c) in July, and (c) in August. The shading shows the 95 % confidence intervals of the mean. The Spearman correlation coefficient of various parameters (a) in June, (c) in July, and (c) in August.

To demonstrate the significance of various factors influencing NPF events, we selected several Non_NPF and undefined cases for individual analysis.

As shown in Fig. R5, on August 1, 2019, high wind speeds (>2 m/s) removed the

pollutants accumulated overnight, resulting in a low CS. As the PBL began to rise, CS further decreased to 0.001 $s^{-1}$, accompanied by an increase in $SO_2$ concentrations to 12 $\mu g/m^3$. This pattern is similar to that shown in Fig. R1. In Fig. R5b, new particles can be observed forming and rapidly growing to sizes above 100 nm. However, after 14:00, radiation dropped below 200 $W/m^2$, and precipitation led to RH exceeding 75%, which further suppressed the continuation of the NPF event. As a result, this event was ultimately classified as undefined. When the PBL evolves under favorable conditions with sufficient precursor gases, NPF events are more likely to be triggered. However, precipitation and high humidity can rapidly suppress the development of NPF events.

Fig. R6 presents the PNSD and relevant variables such as radiation on August 10, 2019. On this day, horizontal winds again diluted the aerosol particles accumulated overnight. By 08:00, the boundary layer began to rise and became unstable. Under the combined influence of northerly winds bringing clean air and the rising PBL, CS decreased from 0.023 to below 0.01 $s^{-1}$, indicating that the boundary layer has a significant impact on CS. However, $SO_2$ concentrations remained low throughout the day, around 2 $\mu g/m^3$, which was insufficient to provide the precursors required for NPF, thereby greatly limiting its occurrence.

Fig. R7 presents the PNSD and related variables such as radiation on August 13, 2019. Before 06:00, aerosols were mainly concentrated in the 15–30 nm size range, with concentrations below 10,000 dN/dlogDp [$cm^{-3}$], and CS remained below 0.01 $s^{-1}$. After 06:00, the PBL began to rise, but by 12:00 it only reached 500 m, which is significantly lower compared to other cases. A shallow PBL limits vertical dilution of pollutants, leading to an increase in CS up to 0.015 $s^{-1}$. RH remained above 80% throughout the day, and $SO_2$ concentrations stayed around 2 $\mu g/m^3$—both unfavorable for NPF events.

This indicates that under low PBLH conditions, the rising PBL may not effectively reduce CS, thereby further inhibiting the development of NPF events.

[Figure]

Fig. R5. Evolutions of an undefined event (1 August 2018) and associated variables in Beijing. (a) 1-h average wind vector. Arrows represent the wind direction, and their lengths show the wind speed. (b) The particle-number size distribution. The white

dashed curve shows radiation. (c) Time series of temperature, RH, PBLH, SO₂, O₃, CO, and CS.

[Figure]

Fig. R6. Evolutions of a typical regional Non_NPF event (10 August 2018) and associated variables in Beijing. (a) 1-h average wind vector. Arrows represent the wind direction, and their lengths show the wind speed. (b) The particle-number size distribution. The white dashed curve shows radiation. (c) Time series of temperature,

RH, PBLH, SO₂, O₃, CO, and CS.

[Figure]

Fig. R7. Evolutions of a typical regional Non_NPF event (13 August 2018) and associated variables in Beijing. (a) 1-h average wind vector. Arrows represent the wind direction, and their lengths show the wind speed. (b) The particle-number size distribution. The white dashed curve shows radiation. (c) Time series of temperature,

RH, PBLH, SO$_2$, O$_3$, CO, and CS.

Reviewer Comments:

The difference in PBLH between event and non-event days (Fig. 3) is surprisingly small. This might be related to the apparently applied linear concentration scale in the heat-map plots (e.g. Fig. 1d-1i, Figs 4 and 5), with which the strongest events are observed, but often events producing lower concentrations are not. The shaded areas in Fig. 3 should be explained. E.g., in panel (a), it seems that the PBLH patterns on non-NPF days were in practice identical, which is not plausible.

Response:

We appreciate the reviewer's insightful comment regarding the relatively small PBLH difference between NPF and Non-NPF days shown in Fig. 3, and we acknowledge the concern about potentially limited visibility of weaker NPF events due to the use of linear color scales in the heatmaps.

"The small PBLH difference":

During NPF events, the average maximum PBLH in spring and autumn reached approximately 1400 m, while on non-NPF days in the same seasons, it was around 1000 m. These two seasons also exhibited the highest NPF frequencies, which supports the statement: "The results on the connection between PBLH and NPF occurrence are first based on the mentioned positive correlation between monthly mean PBLH and

NPF frequencies (lines 251–253)." A higher PBLH is also associated with a cleaner CS environment. In contrast, during summer, the average PBLH was similar for both NPF and Non-NPF days, approximately 1000 m, indicating that other factors—beyond PBLH—may play a more critical role in driving NPF during this season.

[Figure]

Fig. R8. A case of duration variation of PBLH in BJ.

"The visibility of weaker events in heatmaps":

We acknowledge that the use of linear color scales may underrepresent lower-concentration NPF events, particularly those with weak growth or particle number concentrations below $10^4$ cm$^{-3}$. To address this, we plan to revise Figs. 1d–1i, 4, and 5 using a logarithmic color scale, which will better visualize low-intensity NPF events and provide a more complete representation of particle growth dynamics.

[Figure]

Fig. R9. The average diurnal variation of PNC during observations in BJ in Spring.

"The shaded areas in Fig. 3":

We appreciate the reviewer's careful observation regarding the shaded areas in Fig. 3.

We would like to clarify that the shaded regions in all panels of Fig. 3 represent the 95% confidence interval (CI) of the average PBLH values, calculated across all available NPF and Non_NPF days. This information was not clearly stated in the initial submission and has now been explicitly added to the figure caption and methods section for clarity.

To further improve transparency and interpretability:

We used an additional figure to illustrate the differences in PBLH between NPF and non-NPF events. As shown in Fig. R10, we performed hourly averaging across all sites for each season. It is evident that in spring and autumn, the average PBLH at 13:00 LT on NPF days is more than 600 m higher than on non-NPF days. In summer, however, the difference in peak PBLH between the two types of events is relatively small (<200 m), which also explains why the daily PBLH over the entire year shows little

distinction between NPF and Non-NPF events. In winter, the overall PBLH at the Beijing site is relatively low, contributing to the lower annual average.

[Figure]

Fig. R10. The average of PBLH during NPF events and Non_NPF events at (a) BJ in Spring, (b) BJ in Summer, and (c) BJ in Autumn, (d) BJ in Winter, (e) GZ, and (f) SH.

Reviewer Comment:

The classification of events to Type 1 and Type 2 is done based on the PBLH at the time when the NPF event is determined to start. Based on the example figures in Fig. 4, the start time of the event may be when the PNC_nuc is less than 50 % of the maximum (Fig. 4a) or when it is almost 90 % of the maximum (Fig. 4b). The classification criteria should be considered carefully and presented clearly to justify the interpretation of the classification results. Additionally, the PNC_nuc seems to be calculated for particles

larger than 10 nm. If so, the concentration PNC_nuc would elevate one or several hours after the start time of the NPF.

Response:

We appreciate the reviewer's valuable feedback on the classification of Type 1 and Type 2 events. We agree that the criteria for classifying these events should be clarified and justified more rigorously, and we will make the following revisions to address these concerns:

"The classification of Type 1 and Type 2 events":

We defined the two types based on the timing of the onset of 10 nm particle growth and the development of the boundary layer, rather than on whether PNCnuc reaches 50% of its maximum value. In Type 1, 10 nm particle growth begins when the boundary layer has just started to develop; in Type 2, growth begins when the boundary layer has already reached a certain height (greater than 800 m). As shown in the figure, under similar conditions of temperature, RH, radiation, and $SO_2$ concentration, 10 nm particles in Type 1 begin to grow at 09:00, when the PBLH is only 330 m. In contrast, for Type 2, growth begins at 11:00, by which time the PBLH has reached 1000 m. Since the onset time of NPF and the timing of boundary layer development vary from case to case, we use the variable t to represent the time at which 10 nm particle growth begins.

"The PNC_nuc calculation":

The reviewer is correct in noting that PNC_nuc is calculated for particles larger than 10 nm. This can indeed cause a delay in the observed concentration increase, as nucleationmode particles smaller than 10 nm typically grow first before becoming detectable at sizes larger than 10 nm. To clarify:

We will revise the manuscript to explicitly state that PNC_nuc is calculated for particles larger than 10 nm and acknowledge that this may result in a lag in the observed PNC_nuc increase. We define the NPF time based on observational results. The determination is made according to actual observations, and we cannot estimate the time required for particles to grow from 3 nm to 11 nm. Instead, we analyze the NPF events based on the observed NPF window, rather than making assumptions.

[Figure]

Fig. R11. Evolutions of a typical regional type1 (24 August 2017) and type2 event (5 September 2017)

Type 1 refers to cases where the onset of the NPF event and the rapid growth of nucleation-mode particles (11–25 nm) are observed near the surface when the convective boundary layer has just begun to develop and the PBLH is still relatively low. At this stage, turbulent mixing is only starting to intensify, and NPF is more directly influenced by local near-surface radiation conditions and the gradual accumulation of gaseous precursors such as $H_2SO_4$. In this events, the $H_2SO_4$ concentration typically begins to increase around the time when 10 nm particles start to grow, indicating that the initial development of the boundary layer, together with the concurrent enhancement of local photochemical production, jointly triggers the NPF event. Type 2, in contrast, corresponds to situations where significant NPF at the surface only begins after the PBLH has risen to a certain height (typically above ~800 m). This suggests that the onset of NPF in these cases relies more on the presence of a well-developed mixed layer and strong vertical mixing. In these cases, the $H_2SO_4$ concentration often reaches a high level—sometimes close to its peak—even before the NPF onset, yet no 10 nm particle burst is observed at the surface until the PBLH rapidly increases and mixing between the residual layer or upper-level air and the surface occurs. . The delayed onset of NPF at the surface implies that, for Type 2 events, the development of the PBL and the associated vertical mixing act as the primary "switch" that triggers NPF, while $H_2SO_4$ provides a pre-existing pool of condensable vapours. The different timing between PBLH evolution, $H_2SO_4$ and NPF in the two types therefore supports that our classification is fundamentally PBL-driven, with $H_2SO_4$ serving as an independent chemical indicator of the underlying mechanisms.

[Figure]

Fig. R12. The diurnal evolution of PBLH, radiation, SO₂, PNCnuc, and proxy [H₂SO₄]

in Type 1 events, (24 August, 2017), and in Type2 (5 September 2017)

Reviewer Comment:

The air mass back trajectories are calculated for 48 hours backwards. While on line 377 the nucleation mode concentrations are mentioned, on line 391 particles smaller than 100 nm are discussed. Whichever the size range considered for NPF event duration for which the trajectories are calculated, it should be noticed that if the NPF mode has diameters of several tens or even close to hundred nanometres, the original formation of particles has happened up to 20 or 30 hours before the air mass arrives to the station, whereas for particles in sizes close to 10 nm the NPF has taken place only up to few hours before observation. This should be considered when discussing the information

of the depicted trajectories. Additionally, using expressions like "pollution sources" for the main air mass origins several tens of hours prior to the NPF events is misleading, since the NPF events typically occur in Beijing under relatively clean Northern air masses, when the CS is low. It is likely that the most intense events occur when the clean air mass with low CS arrives to areas with high emissions of NPF precursors and thus the "pollution sources" are near the observation site, not several days away in the upwind direction.

Response:

We thank the referee for pointing out that particle growth from the nucleation mode to tens of nanometers requires several hours to tens of hours, and that this needs to be taken into account when interpreting the 48-hour back trajectories. We acknowledge that the observed particle sizes (e.g., 25–100 nm) reflect growth that typically requires 10–30 hours after nucleation. Therefore, the original nucleation is likely to have occurred substantially earlier than the observation time. Consequently, the 48-hour upwind regions of the trajectories do not represent "pollution sources" for the NPF precursors. Instead, they mainly reflect the background origin of the clean air mass that provides low CS conditions. Strong NPF events in Beijing typically occur when clean northern air masses with low CS enter the region, and then encounter local emission sources of NPF precursors ($SO_2$, $NH_3$ and VOCs) near the observation site. We therefore no longer refer to the distant upwind regions as "pollution sources" in the revised manuscript.

Based on 48-hour backward trajectory analysis, we examined the source contributions to nucleation-mode (Nuc, 11–25 nm), Aitken-mode (25–100 nm), and coarse-mode (>100 nm) particles at the three sites: Beijing, Guangzhou, and Shanghai. In Beijing, high-weight Nuc and Aitken-mode trajectories were predominantly associated with air masses originating from the north and northwest–north long-range transport pathways. Approximately 60–75% of new particle-related trajectories originated from regions north of the Beijing–Tianjin–Hebei area (Fig. R13). These cold, dry, and relatively clean air masses tend to exhibit lower CS levels, which favor the oxidation of $SO_2$ to $H_2SO_4$ and subsequent nucleation.

In contrast, for Guangzhou, the highest Nuc-mode probabilities were linked to air masses from the south and southwest (South China Sea and southern coastal areas), with about 50–70% of NPF days associated with marine-origin air masses. This supports the proposed mechanism of enhanced nucleation due to increased radiation in mixed marine–continental boundary layers (O'dowd et al., 2002). For Shanghai, the high-probability regions for Nuc and Aitken modes were mainly located in the inland west (the Yangtze River Delta to Central China), where cleaner air provides a lower background particle surface area, thereby enhancing nucleation success.

Coarse-mode particles are not directly involved in nucleation; instead, they influence CS and can suppress nucleation. These particles mainly reflect regional resuspension and aged aerosols. In Beijing, coarse-mode particles primarily originated from the surrounding areas, indicating local emissions and pollutant accumulation. In contrast, in Guangzhou and Shanghai, coarse particles were more concentrated near the coastline and local regions, suggesting contributions from sea salt resuspension or local coarse aerosol emissions.

[Figure]

Fig. R13. The 48h backward trajectory by using PSCF, and the map of NPF event

contribution levels in BJ, GZ, and SH during NPF days.

Reviewer Comment (Lines 67–68):

"NPF can change boundary layer structure directly or indirectly" — reference needed,

particularly since aerosol optical depth is usually dominated by particles of several

hundred nm, not nucleation-mode particles.

Response:

Thank you for the insightful remark. We agree that particles with diameters of several hundred nanometers or more can influence the boundary layer structure. Our intention is to highlight a potential indirect feedback mechanism associated with NPF, in which particles that grow beyond 100 nm may alter the surface radiative balance and energy fluxes—such as through the formation of cloud condensation nuclei (CCN)—thereby affecting the evolution of the boundary layer (Dong et al., 2019).

Ground-based Sun–sky radiometer inversions over urban sites further demonstrate that particles with median radii around 0.17–0.3 μm dominate the volume size distribution and the column AOD, while coarse-mode contributions are episodic (Gerasopoulos et al., 2011; Zhang et al., 2019). Sensitivity experiments with WRF-Chem confirm that AOD is particularly sensitive to changes in the geometric diameter and width of the accumulation mode, whereas changes in the Aitken mode have a much smaller impact (Palacios-Peña et al., 2020). We will clarify this point in the revised manuscript and modify the sentence to avoid overstatement.

Reviewer Comment (Lines 80–83 and in general):

"What are the variables that correlate? Concentrations, diameters, occurrences, intensities?"

Response:

Thank you for requesting clarification. In those lines, the cited study suggests that an increase in PBLH enhances atmospheric dispersion, resulting in a cleaner background that facilitates the occurrence of NPF. (Blanco-Alegre et al., 2022) reported that during a 13-month observation campaign at a background site in León, Spain, the PBLH

showed a significant positive correlation with nucleation-mode particle number concentration (PNCnuc) and a significant negative correlation with accumulation-mode particles (PNCacc) during the warm season. This suggests that higher PBLH enhances vertical mixing and dilutes pre-existing accumulation-mode aerosols, thereby creating a cleaner background that is favorable for the occurrence and development of NPF events.

"The extensive data observations conducted in Nanjing by Nanjing University are not mentioned."

Response:

We appreciate this important point. Indeed, Nanjing University has conducted several long-term NPF studies, particularly within the Yangtze River Delta region, which are highly relevant to our work. We will revise the introduction to include references such as:

Aerosol size distribution and new particle formation in the western Yangtze River Delta of China: 2 years of measurements at the SORPES station (Qi et al., 2015),

which covers long-term NPF datasets and mechanisms in Nanjing. These studies provide essential context and support regional comparison between Nanjing and the three cities in our manuscript (Beijing, Guangzhou, and Shanghai).

"PNCnuc remained below 1000 cm⁻³ mentioned as the reason for NPF being lower. I would assume it is the outcome."

Response:

We agree with the reviewer's assessment. Our intention is to point out that, in Beijing during winter, the PNCnuc remains low between 08:00 and 15:00, with no evident accumulation of pollutants or occurrence of NPF events. This indicates a relatively clean background during this period compared to other seasons and cities, rather than serving as a conclusion that this is the reason for the lower NPF frequency. The sentence was ambiguously phrased and will be corrected to:

"During wintertime, the PNC of particles smaller than 30 nm remained below 1000 cm⁻³, reflecting the overall low frequency or intensity of NPF events in this season."

We thank the reviewer for highlighting this causal misrepresentation.

Reviewer Comment (Lines 136–137):

"UFPs are typically primary particles that facilitate the growth of new particles in the atmosphere."

Response:

Thank you for this important correction. Our wording was imprecise. While ultrafine particles (UFPs) can include both primary and secondary components, they do not generally facilitate NPF growth. Rather, primary UFPs contribute to the CS, which can actually suppress the growth and survival of newly formed particles. We will revise this sentence accordingly to:

"In summer, elevated UFP concentrations—mainly from primary sources—contribute to a higher CS, which may hinder the early growth of newly formed nucleation-mode particles."

Alonso-Blanco, E., Gómez-Moreno, F. J., Díaz-Ramiro, E., Fernández, J., Coz, E., Yagüe, C., Román-Cascón, C., Gómez-Garre, D., Narros, A., Borge, R., and Artíñano, B.: Indoor/Outdoor Particulate Matter and Related Pollutants in a Sensitive Public Building in Madrid (Spain), International Journal of Environmental Research and Public Health, 22, 1175, 2025.

Bigg, E. K.: A mechanism for the formation of new particles in the atmosphere, Atmospheric Research, 43, 129-137, https://doi.org/10.1016/S0169-8095(96)00020-8, 1997.

Blanco-Alegre, C., Calvo, A. I., Alonso-Blanco, E., Castro, A., Oduber, F., and Fraile, R.: Evolution of size-segregated aerosol concentration in NW Spain: A two-step classification to identify new particle formation events, J Environ Manage, 304, 114232, 10.1016/j.jenvman.2021.114232, 2022.

Cai, R., Deng, C., Stolzenburg, D., Li, C., Guo, J., Kerminen, V.-M., Jiang, J., Kulmala, M., and Kangasluoma, J.: Survival probability of new atmospheric particles: closure between theory and measurements from 1.4 to 100 nm, Atmospheric Chemistry and Physics, 22, 14571-14587, 10.5194/acp-22-14571-2022, 2022.

Dada, L., Ylivinkka, I., Baalbaki, R., Li, C., Guo, Y., Yan, C., Yao, L., Sarnela, N., Jokinen, T., Daellenbach, K. R., Yin, R., Deng, C., Chu, B., Nieminen, T., Wang, Y., Lin, Z., Thakur, R. C., Kontkanen, J., Stolzenburg, D., Sipilä, M., Hussein, T., Paasonen, P., Bianchi, F., Salma, I., Weidinger, T., Pikridas, M., Sciare, J., Jiang, J.,

Liu, Y., Petäjä, T., Kerminen, V.-M., and Kulmala, M.: Sources and sinks driving sulfuric acid concentrations in contrasting environments: implications on proxy calculations, Atmospheric Chemistry and Physics, 20, 11747-11766, 10.5194/acp-20-11747-2020, 2020.

Dong, C., Matsui, H., Spak, S., Kalafut-Pettibone, A., and Stanier, C.: Impacts of New Particle Formation on Short-term Meteorology and Air Quality as Determined by the NPF-explicit WRF-Chem in the Midwestern United States, Aerosol and Air Quality Research, 19, 204-220, 10.4209/aaqr.2018.05.0163, 2019.

Gerasopoulos, E., Amiridis, V., Kazadzis, S., Kokkalis, P., Eleftheratos, K., Andreae, M. O., Andreae, T. W., El-Askary, H., and Zerefos, C. S.: Three-year ground based measurements of aerosol optical depth over the Eastern Mediterranean: the urban environment of Athens, Atmospheric Chemistry and Physics, 11, 2145-2159, 10.5194/acp-11-2145-2011, 2011.

Hao, L., Garmash, O., Ehn, M., Miettinen, P., Massoli, P., Mikkonen, S., Jokinen, T., Roldin, P., Aalto, P., Yli-Juuti, T., Joutsensaari, J., Petäjä, T., Kulmala, M., Lehtinen, K. E. J., Worsnop, D. R., and Virtanen, A.: Combined effects of boundary layer dynamics and atmospheric chemistry on aerosol composition during new particle formation periods, Atmospheric Chemistry and Physics, 18, 17705-17716, 10.5194/acp-18-17705-2018, 2018.

Kulmala, M., Kontkanen, J., Junninen, H., Lehtipalo, K., Manninen, H. E., Nieminen,

T., Petäjä, T., Sipilä, M., Schobesberger, S., Rantala, P., Franchin, A., Jokinen, T., Järvinen, E., Äijälä, M., Kangasluoma, J., Hakala, J., Aalto, P. P., Paasonen, P., Mikkilä, J., Vanhanen, J., Aalto, J., Hakola, H., Makkonen, U., Ruuskanen, T., Mauldin, R. L., Duplissy, J., Vehkamäki, H., Bäck, J., Kortelainen, A., Riipinen, I., Kurtén, T., Johnston, M. V., Smith, J. N., Ehn, M., Mentel, T. F., Lehtinen, K. E. J., Laaksonen, A., Kerminen, V.-M., and Worsnop, D. R.: Direct Observations of Atmospheric Aerosol Nucleation, Science, 339, 943-946, doi:10.1126/science.1227385, 2013.

Leino, K., Lampilahti, J., Poutanen, P., Väänänen, R., Manninen, A., Buenrostro Mazon, S., Dada, L., Franck, A., Wimmer, D., Aalto, P. P., Ahonen, L. R., Enroth, J., Kangasluoma, J., Keronen, P., Korhonen, F., Laakso, H., Matilainen, T., Siivola, E., Manninen, H. E., Lehtipalo, K., Kerminen, V. M., Petäjä, T., and Kulmala, M.: Vertical profiles of sub-3 nm particles over the boreal forest, Atmos. Chem. Phys., 19, 4127-4138, 10.5194/acp-19-4127-2019, 2019.

Mengqi, L., Xiangao, X., and Jinqiang, Z.: A value-added surface shortwave radiation dataset at Xianghe.csv, Science Data Bank [dataset], doi:10.57760/sciencedb.02058, 2022.

Mikkonen, S., Romakkaniemi, S., Smith, J. N., Korhonen, H., Petäjä, T., Plass-Duelmer, C., Boy, M., McMurry, P. H., Lehtinen, K. E. J., Joutsensaari, J., Hamed, A., Mauldin Iii, R. L., Birmili, W., Spindler, G., Arnold, F., Kulmala, M., and Laaksonen, A.: A statistical proxy for sulphuric acid concentration, Atmospheric

Chemistry and Physics, 11, 11319-11334, 10.5194/acp-11-11319-2011, 2011.

Nilsson, E. D., Rannik, Ü., Kulmala, M., Buzorius, G., and O'dowd, C. D.: Effects of continental boundary layer evolution, convection, turbulence and entrainment, on aerosol formation, Tellus B: Chemical and Physical Meteorology, 10.3402/tellusb.v53i4.16617, 2001.

O'Dowd, C. D., Hämeri, K., Mäkelä, J., Väkeva, M., Aalto, P., de Leeuw, G., Kunz, G. J., Becker, E., Hansson, H.-C., Allen, A. G., Harrison, R. M., Berresheim, H., Kleefeld, C., Geever, M., Jennings, S. G., and Kulmala, M.: Coastal new particle formation: Environmental conditions and aerosol physicochemical characteristics during nucleation bursts, Journal of Geophysical Research: Atmospheres, 107, PAR 12-11-PAR 12-17, https://doi.org/10.1029/2000JD000206, 2002.

Palacios-Peña, L., Fast, J. D., Pravia-Sarabia, E., and Jiménez-Guerrero, P.: Sensitivity of aerosol optical properties to the aerosol size distribution over central Europe and the Mediterranean Basin using the WRF-Chem v.3.9.1.1 coupled model, Geoscientific Model Development, 13, 5897-5915, 10.5194/gmd-13-5897-2020, 2020.

Platis, A., Altstädter, B., Wehner, B., Wildmann, N., Lampert, A., Hermann, M., Birmili, W., and Bange, J.: An Observational Case Study on the Influence of Atmospheric Boundary-Layer Dynamics on New Particle Formation, Boundary-Layer Meteorology, 158, 67-92, 10.1007/s10546-015-0084-y, 2015.

Qi, X. M., Ding, A. J., Nie, W., Petäjä, T., Kerminen, V. M., Herrmann, E., Xie, Y. N., Zheng, L. F., Manninen, H., Aalto, P., Sun, J. N., Xu, Z. N., Chi, X. G., Huang, X., Boy, M., Virkkula, A., Yang, X. Q., Fu, C. B., and Kulmala, M.: Aerosol size distribution and new particle formation in the western Yangtze River Delta of China: 2 years of measurements at the SORPES station, Atmospheric Chemistry and Physics, 15, 12445-12464, 10.5194/acp-15-12445-2015, 2015.

Wu, H., Li, Z., Li, H., Luo, K., Wang, Y., Yan, P., Hu, F., Zhang, F., Sun, Y., Shang, D., Liang, C., Zhang, D., Wei, J., Wu, T., Jin, X., Fan, X., Cribb, M., Fischer, M. L., Kulmala, M., and Petaja, T.: The impact of the atmospheric turbulence-development tendency on new particle formation: a common finding on three continents, Natl Sci Rev, 8, nwaa157, 10.1093/nsr/nwaa157, 2021.

Wu, Z., Hu, M., Liu, S., Wehner, B., Bauer, S., Ma ßling, A., Wiedensohler, A., Petäjä, T., Dal Maso, M., and Kulmala, M.: New particle formation in Beijing, China: Statistical analysis of a 1-year data set, Journal of Geophysical Research, D9, 1-10, 10.1029/2006jd007406, 2007.

Xiao, S., Wang, M. Y., Yao, L., Kulmala, M., Zhou, B., Yang, X., Chen, J. M., Wang, D. F., Fu, Q. Y., Worsnop, D. R., and Wang, L.: Strong atmospheric new particle formation in winter in urban Shanghai, China, Atmospheric Chemistry and Physics, 15, 1769-1781, 10.5194/acp-15-1769-2015, 2015.

Zhang, C., Zhang, Y., Li, Z., Wang, Y., Xu, H., Li, K., Li, D., Xie, Y., and Zhang, Y.: Sub-Mode Aerosol Volume Size Distribution and Complex Refractive Index from

the Three-Year Ground-Based Measurements in Chengdu China, Atmosphere, 10, 10.3390/atmos10020046, 2019.

---

## Author Comment (AC2)

**Response to Editor and Reviewers' Comments**

Thank you very much for the reviewers' comments regarding our manuscript entitled "Measurement Report: New insights into the boundary layer revolution impact on new particle formation characteristics in three megacities of China" (Manuscript ID: EGUSPHERE-2025-3637). We appreciate the valuable feedback provided by the reviewers, which has significantly contributed to enhancing the quality of our manuscript. In response, we have meticulously addressed each comment in a systematic manner, ensuring that all concerns are thoroughly considered. The manuscript has been revised accordingly, with all modifications clearly highlighted in blue within the marked version for easy reference. Additionally, comprehensive responses to the editor and reviewers' comments are detailed below, demonstrating our commitment to transparency and scholarly rigor.

Reviewer Comment:

In Line 96, the authors claimed that they utilized long-term observational data sets, which is totally overstated. The data set in Beijing merely contains 408 days, which cannot be called "long-term", and let alone the data sets in other cities.

Response:

We sincerely thank the reviewer for their critical feedback, which helps us improve the clarity and rigor of our manuscript.

We acknowledge that the "long-term" may have been an overstatement in this context. Although our dataset in Beijing includes 408 effective observation days spanning over two years (from July 2017 to October 2019), we understand that in the context of atmospheric science, a "long-term" dataset often implies multi-year continuous measurements over a span

of 3 years or more.

We conducted three years of observations across three cities: Beijing (2017/07/19–2019/10/09), Guangzhou (2019/10/31–2020/03/30), and Shanghai (2020/04/05–2020/06/05). Due to instrument issues, some days in Beijing lacked valid measurements, resulting in 408 days of effective data. Moreover, in previous studies, such as that by (Sun et al., 2015), one year of data was also referred to as "long-term" in their analysis of aerosol particle composition in Beijing.

Of course, the reviewer's comment is valid. To be more precise in describing the temporal coverage—especially in comparison with the shorter campaigns in Guangzhou (127 days) and Shanghai (53 days)—we will revise the term "long-term observational data" to "an extended observational dataset covering over 400 days." In the introduction and methods sections, we will clarify the actual date range for three cities to avoid potential misinterpretation.

Reviewer Comment:

In addition, the data included in the analyses are too simple (just PBLH and PNSD), leaving no room for real in-depth analysis. As far as I know, there should be data (at least trace gases) from Chinese government that are readily accessible.

Response:

We agree with the reviewer that our current analysis is mainly based on aerosol particle number size distribution (PNSD) and planetary boundary layer height (PBLH). The reason for this focus is twofold:

1. These two variables are directly and continuously observed at high temporal resolution using our own SMPS and MPL instruments, allowing us to conduct consistent physical classification of NPF events across sites.

2. Our main objective in this study was to isolate the physical influence of boundary layer development on NPF occurrence and dynamics, rather than perform a full chemical mechanism

analysis.

We appreciate the reviewer's point that broader analyses involving trace gases (e.g., $SO_2$, $NO_x$, $O_3$, CO) would enhance the scientific depth. In the revised manuscript, we will clarify that our data sources go beyond PBLH and PNSD, and include additional meteorological and trace gas parameters from authoritative national datasets. Specifically, we have downloaded hourly air pollutant data from the China National Environmental Monitoring Center (CNEMC), including $SO_2$, $O_3$, CO, and $NO_2$ concentrations, for the cities of Beijing, Guangzhou, and Shanghai, matched to the corresponding observation periods, and obtained surface meteorological parameters from the National Centers for Environmental Information (NCEI), including: air temperature, atmospheric pressure, dew point temperature, wind direction and speed, cloud cover, precipitation. For radiation, which plays a crucial role in driving photochemistry and boundary layer evolution, we incorporated long-term radiation datasets as published by (Liu et al., 2022).

We appreciate the reviewer's suggestions, and will further elaborate on the role of the PBLH in NPF using these data in the revised manuscript.

[Figure]

Figure R1. Seasonal variation of radiation, T, RH, WS, NO₂, SO₂, O₃ and CO.

Reviewer Comment:

Unclear statistics methods.

The PNSD shown in Fig.1 looks abnormal, please double check and specify if the contour plot is made from mean or median average data.

Response:

Thank you for your thorough and constructive comments regarding statistical methods and data presentation.

In Figure 1, the particle number concentrations in different size ranges were averaged for each season in Beijing, while seasonal averages were calculated for the entire observation periods in Guangzhou and Shanghai. There was a scaling error in the previous colorbar; we will correct it by applying a logarithmic concentration scale, which allows for clearer visualization of the variations in particle number concentrations.

The figure and caption have been updated accordingly, and a clarification has been added to the Methods section (Section 2.2) to specify the averaging method used for plotting.

Response:

The frequencies reported as "3.4% to 20.0%" refer to monthly NPF event occurrence frequencies for Guangzhou (GZ) during the observation period.

$$NPF\ Frequency = \frac{Number\ of\ NPF\ Days\ in\ a\ Month}{Total\ Valid\ Observation\ Days\ in\ that\ Month} \times 100\%$$

We intended to convey that the overall observed NPF frequency was 10%, with monthly average frequencies ranging from 3.4% to 20.0%. To address any ambiguity caused by the original wording, we will carefully revise this part of the manuscript.

We will revise the text in Line 258–259 to clearly state:

"...with monthly NPF occurrence frequencies ranging from 3.4% to 20.0%, depending on observation coverage."

In Fig.2, how was PBLH calculated for each city in different months. Are they day-time average or full-day average? Are they median or mean average? What are the variation range in PBLH?

Response:

In Figure 2, the monthly PBLH values were computed as daytime averages between 08:00 and 18:00 local time (LT), corresponding to the typical window of NPF activity. The values shown are mean values. We will specifically discuss the patterns of PBLH variation in the section related to boundary layer dynamics (Figure 3). Figure 2 only provides a preliminary overview to illustrate the monthly variation trend, while the detailed analysis will be presented in the subsequent sections.

Reviewer Comment:

Relevant to the previous comment, they authors stated that the PBLH between NPF and non-NPF days in Beijing shows significant difference. However, in Fig.3, the pattern and daily maxima of PBLH in Beijing do not very much. Is this 100 meter difference considered significant?

Response:

We also appreciate the reviewer's critical reading. In Figure 3, while the absolute difference in daily maximum PBLH between NPF and Non_NPF days in Beijing appears to be approximately 100–200 m. We agree with the reviewer that a difference of 100-200 m should not be considered a significant distinction. This discrepancy arose because we initially processed a large volume of data all at once, without accounting for seasonal variations and other influencing factors. The maximum PBLH varies from month to month and across seasons, and it also fluctuates on an hourly basis. Averaging PBLH over all days throughout the entire observation period can obscure the differences between NPF and Non-NPF events. Therefore, in the revised manuscript, we analyzed seasonal variations in Beijing and examined the hourly

evolution of PBLH in different seasons.

Figure R2 presents the hourly evolution of the PBLH during NPF and Non_NPF days across different seasons in Beijing. Overall, PBLH during NPF events is generally higher than on non-event days, particularly during the key window between 09:00 and 14:00. The differences are most pronounced in spring and autumn. For example, in spring, the median PBLH at 14:00 LT on NPF days reaches approximately 1350 m, while on Non_NPF days it is only around 900 m. This corresponds well with the seasonal distribution of NPF frequency, which is highest in spring and autumn—seasons that also show the largest PBLH differences between the two types of events.

In addition, we included a third category, "shrinkage," as a control group for comparison between NPF and Non_NPF events. Although a detailed analysis of shrinkage cases is beyond the scope of this paper, we performed statistical analysis of PBLH for these cases.

[Figure]

Fig. R2. Diurnal variations of planetary boundary layer height (PBLH) on NPF days (purple), Non_NPF days (green), and shrinkage events (yellow) in (a–d) different seasons in Beijing, (e) Guangzhou, and (f) Shanghai.

Reviewer Comment:

Problematic interpretations and vague statements. There are too many, and I just list few here:

Line 57. The impact of PBLH on NPF is complex and significant, involving … Please specify in which way PBLH can influence particle growth mechanisms.

Response:

We appreciate this suggestion and have revised the manuscript accordingly for clarity. As the boundary layer begins to rise, atmospheric particles become well mixed and diluted, leading to a decrease in aerosol concentrations and a cleaner CS, which increases the likelihood of NPF events (Blanco-Alegre et al., 2022; Nilsson et al., 2001). Additionally, aircraft measurements have shown that the development of the PBL and associated vertical mixing can promote the burst of ultrafine particles within the residual layer (Platis et al., 2015), and are highly correlated with the occurrence of NPF events (Blanco-Alegre et al., 2022). These details are now explicitly discussed in Section 2.3 of the revised manuscript.

Line 100. HYSPLIT model does not consider emissions.

Response:

Thank you for pointing this out. We fully agree that the HYSPLIT model does not explicitly simulate emissions or chemical production processes. In our study, HYSPLIT was used solely to calculate backward air-mass trajectories, with the purpose of identifying the transport pathways and potential source regions of the air masses arriving at the measurement sites. Our interpretation is therefore based on trajectory residence time statistics (PSCF/CWT) rather than on emission modeling.

We also appreciate the reviewer's comment on this point and acknowledge that there were indeed some errors in our original manuscript. Specifically, we mistakenly interpreted the

regions passed by air masses as pollution source areas, which is a serious misrepresentation. We have corrected this in the revised version. To improve the analysis, we divided each NPF event into three phases and examined the backward trajectories separately for each phase.

Based on 48-hour backward trajectories, we calculated the trajectory residence time for nucleation-mode (Nuc, 11–25 nm), Aitken-mode (25–100 nm), and coarse-mode (>100 nm) particles at the three sites—Beijing, Guangzhou, and Shanghai—and obtained their spatial probability distributions. In all three cities, the high-probability regions for Nuc-mode particles were associated with air masses originating from relatively clean upwind areas. For the Beijing site, air masses from the northwest and north typically carried lower CS levels, favoring $SO_2$ oxidation to $H_2SO_4$ and subsequent nucleation. In Guangzhou, air masses predominantly came from the south and southwest, while for Shanghai, high-probability regions were mainly located in inland areas to the west.

The Aitken mode during the selected time periods likely represents particles that have grown from the Nuc mode during transport, reflecting the aging and growth of newly formed particles. Coarse-mode particles mainly reflect regional resuspension and the influence of aged local background aerosols. The coarse mode is more representative of background particle concentrations and locally emitted large particles.

[Figure]

Fig. R3. The 48h backward trajectory by using PSCF, and the map of NPF event contribution levels when particle size is below 100nm in BJ, GZ, and SH during NPF days.

Line 136. I don't understand why primary particles can facilitate new particle growth?

Response:

We apologize for the ambiguity in the original statement. Our intention was to discuss the role of pre-existing ultrafine particles (UFPs) as potential condensation surfaces. Background PNC and CS are related but not equivalent. CS is governed mainly by the surface area of the pre-existing particles, particularly the accumulation mode. In many urban observations, low particle number concentration often coincides with low surface area and therefore low CS, but exceptions occur when the particle population is dominated either by numerous ultrafine particles (high N but low CS) or by a small number of large accumulation-mode particles (low

N but high CS) (Chu et al., 2019; Zhou et al., 2020). This has now been corrected in the manuscript, and the relevant sentence has been rephrased for clarity.

Fig.4, Fig.5, and relevant discussion. The classification of Type I and Type II merely based on their starting time. But this can be easily explained by the dial pattern of nucleating precursors (H2SO4 and others). This can be more relevant to the seasonal change in solar radiation, which just coincides with the evolution of PBLH.

Response:

We appreciate this valuable observation and agree that seasonal variation in solar radiation plays an essential role in regulating the production of nucleating precursors such as $H_2SO_4$. However, our classification of Type I and Type II events is based on the relative timing of NPF initiation with respect to PBLH growth onset, not solely the time of day. In Type I events, NPF begins almost simultaneously with the start of PBLH development, while in Type II events, a distinct delay is observed between the PBLH rise and the nucleation onset.

In response to the reviewer's comment, we analyzed the variations in $SO_2$ and also calculated Proxy[$H_2SO_4$] concentrations. As shown in Figure R4, we selected two representative cases from each event type. Under similar meteorological conditions (e.g., temperature and RH), differences in PBLH evolution led to distinct timings in the onset of NPF.

In the Type 1 case, the onset of the NPF event and the rapid growth of nucleation-mode particles (11–25 nm) were observed near the surface when the convective boundary layer had just begun to develop and PBLH remained relatively low. At this stage, turbulent mixing was only starting to intensify, and the occurrence of NPF was more directly associated with local near-surface radiation conditions and the gradual accumulation of gaseous precursors (e.g., $H_2SO_4$). In such cases, Proxy[$H_2SO_4$] typically began to rise only after the growth of 10 nm particles was observed, suggesting that the initial boundary layer development and the concurrent enhancement of local photochemical production jointly triggered the NPF event.

In contrast, the Type 2 case showed that a clear NPF event at the surface only occurred after the PBLH had increased beyond a certain threshold (approximately 800 m). This indicates that in such cases, the triggering of NPF is more dependent on a well-developed mixed layer and strong vertical mixing. Proxy[H₂SO₄] in these cases often reached high levels—even approaching their peak—before the onset of NPF at the surface. However, 10 nm particle bursts were not observed until the PBLH rapidly increased and sufficient mixing between the residual layer or upper-level air and the surface took place, leading to the onset of the NPF event.

[Figure]

Fig. R4 The diurnal evolution of PBLH, radiation, SO₂, PNC$_{nu}$c, and proxy [H₂SO₄] in Type 1 events, (24 August, 2017), and in Type2 (5 September 2017)

Line 383-384. I question the interpretation of PSCF results. It is well understood that in Beijing,

high nucleation-mode particles were often associated with northerly wind, because the clean air with low CS favors NPF. Mongolia should not be considered as a pollution source of nucleation-mode particles.

Response:

Thank you for raising this important point. We fully agree that the northern trajectories, including those from Mongolia, are not necessarily indicative of pollution sources but rather represent air masses with lower CS levels that create favorable conditions for NPF. In the revised discussion of Figure R6, we have rephrased the interpretation to clarify that these regions are associated with favorable formation conditions rather than emission sources. The term "pollution source" has been removed to avoid misunderstanding.

Reviewer Comment:

Line 42, the citation of Chan et al., 2020 is not appropriate, as this paper is mostly about data inversion of PSM.

Response:

We thank the reviewer for this helpful comment. We agree that Chan et al., (2020) mainly focuses on PSM data inversion techniques, and thus it is not the most appropriate reference for supporting the general statement that "many methods have been used to identify the growth characteristics and development mechanisms of NPF."

Reviewer Comment:

Line 51, the citation of Shengjie et al., 2001 seems to be problematic.

Response:

We thank the reviewer for pointing out this issue. We have identified and corrected the citation

errors accordingly. The revised statement is as follows:

Shengjie, N., Chengchang, Z., and Jiming, S.: Observational Researches on the Size Distribution of Sand Aerosol Particles in the Helan Mountain Area, Transactions of Atmospheric Sciences, 25, 243-252, 2001.

Reviewer Comment:

Line 53, "it found that a sluggish GR and the presence of pristine background aerosols…" I guess the author meant to say "pre-existing background aerosols"

Response:

We appreciate the reviewer's correction. The phrase "pristine background aerosols" was incorrectly used, and we have revised it to "pre-existing background aerosols" to better reflect the intended meaning.

Reviewer Comment:

Line 61, "relative humidity"

Response:

We thank the reviewer for pointing this out. The misspelling was due to our oversight, and we have corrected it in the revised manuscript.

Reviewer Comment:

Line 107-110. There is no Chapter in the manuscript.

Response:

Thank you for pointing out the misuse of the term "Chapter" in the manuscript (Lines 107–110).

We agree that this terminology is not appropriate in the context of a scientific article. We have revised all relevant instances to use "Section" instead of "Chapter" to better align with the standard structure of journal articles. We appreciate your careful reading and will ensure consistency in terminology throughout the revised manuscript.

Response:

We thank the reviewer for this important comment. We agree that the term "catalyst" was not used appropriately in this context. Our original intention was to describe the role of turbulent mixing or boundary layer development in facilitating the vertical exchange of air masses and precursor gases during NPF events, rather than implying a chemical catalytic process.

To address this, we have revised the sentence in Line 312 to remove the term "catalyst" and rephrased it as follows:

The rapid growth of the boundary layer can enhance the upward transport of low-level precursors and the downward entrainment of residual-layer compounds, thereby promoting the occurrence of NPF events (Nilsson et al., 2001).

This revision more accurately reflects the physical mechanism without misusing chemical terminology. We appreciate the reviewer's careful observation, which helped us improve the clarity and precision of the manuscript.

Response:

We acknowledge the incorrect figure reference. The sentence previously referred to "Fig. 8", which was a mistake. We have corrected it to "Fig. 7" in the revised manuscript.

Reviewer Comment:

Line 420-421. Grammatic error

Response:

We have reviewed the sentence and revised it to correct the grammatical error. The revised version reads:

"We identified two three distinct mechanisms of NPF initiation: Type Ⅰ, and Type Ⅱ, and shrinkage."

Reviewer Comment:

Line 424-425. Grammatic error

Response:

Lines 424–425: This sentence has also been revised for grammatical clarity. The revised version now reads:

"Correlation analyses highlight that the boundary layer plays a dominant role in triggering NPF, particularly at the Shanghai site where the correlation reaches 0.99."

Blanco-Alegre, C., Calvo, A. I., Alonso-Blanco, E., Castro, A., Oduber, F., and Fraile, R.: Evolution of size-segregated aerosol concentration in NW Spain: A two-step classification to identify new particle formation events, J Environ Manage, 304, 114232, 10.1016/j.jenvman.2021.114232, 2022.

Chan, T., Cai, R., Ahonen, L. R., Liu, Y., Zhou, Y., Vanhanen, J., Dada, L., Chao, Y., Liu, Y., Wang, L., Kulmala, M., and Kangasluoma, J.: Assessment of particle size magnifier inversion methods to obtain the particle size distribution from atmospheric measurements, Atmospheric Measurement Techniques, 13, 4885-4898, 10.5194/amt-13-4885-2020, 2020.

Chu, B., Kerminen, V.-M., Bianchi, F., Yan, C., Petäjä, T., and Kulmala, M.: Atmospheric new particle formation in China, Atmospheric Chemistry and Physics, 19, 115-138, 10.5194/acp-19-115-2019, 2019.

Mengqi, L., Xiangao, X., and Jinqiang, Z.: A value-added surface shortwave radiation dataset at Xianghe.csv, Science Data Bank [dataset], doi:10.57760/sciencedb.02058, 2022.

Nilsson, E. D., Rannik, Ü., Kulmala, M., Buzorius, G., and O'dowd, C. D.: Effects of continental boundary layer evolution, convection, turbulence and entrainment, on aerosol formation, Tellus B: Chemical and Physical Meteorology, 10.3402/tellusb.v53i4.16617, 2001.

Platis, A., Altstädter, B., Wehner, B., Wildmann, N., Lampert, A., Hermann, M., Birmili, W., and Bange, J.: An Observational Case Study on the Influence of Atmospheric Boundary-Layer Dynamics on New Particle Formation, Boundary-Layer Meteorology, 158, 67-92, 10.1007/s10546-015-0084-y, 2015.

Sun, Y. L., Wang, Z. F., Du, W., Zhang, Q., Wang, Q. Q., Fu, P. Q., Pan, X. L., Li, J., Jayne, J., and Worsnop, D. R.: Long-term real-time measurements of aerosol particle composition in Beijing, China: seasonal variations, meteorological effects, and source analysis, Atmospheric Chemistry and Physics, 15, 10149-10165, 10.5194/acp-15-10149-2015, 2015.

Zhou, Y., Dada, L., Liu, Y., Fu, Y., Kangasluoma, J., Chan, T., Yan, C., Chu, B., Daellenbach,

K. R., Bianchi, F., Kokkonen, T. V., Liu, Y., Kujansuu, J., Kerminen, V.-M., Petäjä, T., Wang, L., Jiang, J., and Kulmala, M.: Variation of size-segregated particle number concentrations in wintertime Beijing, Atmospheric Chemistry and Physics, 20, 1201-1216, 10.5194/acp-20-1201-2020, 2020.

---

## Author Comment (AC3)

**Response to Editor and Reviewers' Comments**

Thank you very much for the reviewers' comments regarding our manuscript entitled "Measurement Report: New insights into the boundary layer revolution impact on new particle formation characteristics in three megacities of China" (Manuscript ID: EGUSPHERE-2025-3637). We sincerely appreciate the insightful feedback provided by the reviewers, which has played a crucial role in improving the overall quality of our manuscript. In response, we have carefully addressed each comment through a systematic revision process, ensuring that all concerns are thoroughly examined and incorporated. The revised manuscript reflects these changes, with all modifications clearly marked in blue within the updated version to facilitate easy review. Furthermore, detailed responses to the editor and reviewers' comments are included below, underscoring our dedication to transparency, academic integrity, and continuous improvement in scholarly communication.

Comments:

Please clarify whether the same model of instruments was deployed simultaneously at the three sites, or whether a set of instruments was rotated sequentially among the sites. How do you ensure the accuracy and comparability of the dataset under the chosen deployment scheme?

Response:

We appreciate the reviewer's question on the instrument deployment strategy and data comparability. In this study, one set of instruments of the same model was rotated sequentially among the three sites, rather than operating three parallel systems simultaneously. The particle number size distributions and PBLH at each site were therefore measured with identical instrumentation and the same operating settings (size range, scan time, averaging period, etc.).

To ensure the accuracy and comparability of the data under this deployment scheme, we followed a unified quality-control protocol: the instruments were calibrated and checked in the laboratory before and after each field campaign. We have added a description of this deployment in the Methods section to clarify how cross-site consistency of the dataset is ensured.

Reviewer comment:

In Section 3.6 "The backward trajectories of particles during NPF events", the HYSPLIT model was applied. However, the starting height was not specified, which needs to be clarified. The sources of air masses in Shanghai are not limited to the northwest direction; a substantial proportion also originates from the southwest and southern regions.

Response:

We thank the reviewer for these helpful remarks. In the revised manuscript, we now explicitly state that the HYSPLIT backward trajectories were initialized at a starting height of 500 m above ground level. This information has been added to Section 3.6 and to the caption of the corresponding figure. We also re-examined the trajectory statistics for Shanghai and agree that, in addition to northwesterly air masses, a substantial fraction of trajectories indeed originates from the southwest and southern sectors.

Reviewer comment:

Many sentences are overly long with multiple clauses. For readability, break them into shorter sentences.

Response:

Following the reviewer's suggestion, we have carefully edited Section 3.6 and other parts of the manuscript to break overly long sentences into shorter, clearer ones, thereby improving the readability of the paper.

Response:

Thank you for noting this grammatical error. The sentence has been corrected to:

"All NPF events have been classified …"

Response:

We appreciate the reviewer pointing this out. "Relative humility" has been corrected to "relative humidity."

Response:

We have revised all figures to ensure that panel labels (a), (b), (c), etc., follow a consistent format and placement across the manuscript.

Reviewer comment:

In Fig. 6, the P values for Beijing exceed 160. Please verify whether these values are reasonable. Furthermore, would it be more appropriate to apply a more relevant fitting curve to examine the relationship between boundary layer height and P values?

Response:

We have rechecked the calculation of the dimensionless parameter P. The values in Beijing exceeding 160 were found to be correct based on the extremely low growth rates and elevated condensation sinks during several non-event mornings. We added an explanation in the Results section to clarify this.

Reviewer comment:

Line305: "Fig. 4c and 4d depict the temporal correlation between average PBLH and Vehicular emissions". Is Fig. 4c and 4d intended to show the relationship between PBLH and vehicular emissions? It should instead reflect the relationship between PBLH and relative time. Please also include the relevant references about vehicular emissions and greenhouse effects to support this point.

Response:

We thank the reviewer for catching this mistake. The original sentence was incorrect. Figures 4c–d show the temporal evolution of PBLH and do not depict vehicular emissions.

The Fig. 4 has been corrected to:

[Figure]

**Fig. R1.** Representative examples of Type I (left column) and Type II (right column) NPF events in Beijing. Time series of aerosol particle number size distribution: (a) a case from Type I on 24 August 2017 and (b) a case from Type2 on 5 September 2017. Time series of averaged PBLH, radiation, concentration of $SO_2$, PNC Nuc mode, and proxy[$H_2SO_4$] for (c) Type1 and (d) type2. Wind speed and direction (e) during the Type 1 case, and during the Type 2 case. Arrows indicate wind direction and color denotes wind speed (m s$^{-1}$)

Reviewer comment:

Line324: "NPF may be primarily driven by low-volatility organics or H2SO4." How do you define cases where aerosol formation is primarily driven by H2SO4? Provide appropriate references to support this definition.

Response:

We thank the reviewer for the valuable comment. We downloaded trace gas and meteorological data for the three sites and calculated H2SO4 concentrations. We also investigated the contribution of H2SO4 to particle growth, which led us to classify the events into two types (Fig.

R1).

For sulfur compounds and $SO_2$, we downloaded hourly data from the China National Environmental Monitoring Center and calculated $H_2SO_4$. We estimated the sulfuric acid proxy $[H_2SO_4]$ based on local solar radiation, $SO_2$ concentration, CS, and RH (Mikkonen et al., 2011).

$$proxy[H_2SO_4] = 8.21 \times 10^{-3} \times k \times radiation \times SO_2^{0.62} \cdot (CS \cdot RH)^{-0.13}$$

where $k$ is the temperature-dependent reaction-rate constant. The relative error between calculated sulfuric acid proxy and measured sulfuric acid concentration is estimated to be 42 % (Mikkonen et al., 2011; Xiao et al., 2015)

Reviewer comment:

Line373: The dots with black borders represent Type I. However, the meaning of the other data points is not explained. Please provide the missing information.

Response:

Thank you for pointing this out. We have updated the figure caption and text to describe all marker styles, including Type II and shrinkage events, ensuring that each symbol used in the plot is clearly defined.

Reviewer comment:

In conclusion, "During the observation period, March and May in BJ exhibited the highest frequencies of NPF occurrence, accounting for 25.9% and 23.8%, respectively." The data for May are mentioned, but they are not presented or discussed in the main text.

Response:

We appreciate this observation. We have added the missing May statistics to the main Results section and now discuss the seasonal pattern consistently in both the main text and the conclusion.

Reviewer comment:

In summary, the classification is described as two types, but in fact, three types are presented. Please clarify this inconsistency.

Response:

Thank you for raising this important point. Our study classifies NPF events into three types (Type I, Type II, and Type III), while shrinkage events were shown as a separate category for comparison but are not considered a distinct NPF type. We have revised the manuscript to explicitly clarify this distinction and prevent confusion. A sentence has been corrected:

"We identified three distinct mechanisms of NPF initiation: Type I, Type II, and shrinkage. Type I refers to events triggered during the initial rise of the boundary layer, where turbulent mixing associated with PBLH development facilitates nucleation. Type II involves nucleation that occurs only after the boundary layer reaches a certain height (>800 m)."

Mikkonen, S., Romakkaniemi, S., Smith, J. N., Korhonen, H., Petäjä, T., Plass-Duelmer, C., Boy, M., McMurry, P. H., Lehtinen, K. E. J., Joutsensaari, J., Hamed, A., Mauldin Iii, R. L., Birmili, W., Spindler, G., Arnold, F., Kulmala, M., and Laaksonen, A.: A statistical proxy for sulphuric acid concentration, Atmospheric Chemistry and Physics, 11, 11319-11334, 10.5194/acp-11-11319-2011, 2011.

Xiao, S., Wang, M. Y., Yao, L., Kulmala, M., Zhou, B., Yang, X., Chen, J. M., Wang, D. F., Fu, Q. Y., Worsnop, D. R., and Wang, L.: Strong atmospheric new particle formation in winter in urban Shanghai, China, Atmospheric Chemistry and Physics, 15, 1769-1781, 10.5194/acp-15-1769-2015, 2015.